# GLP-1 and the Degenerating Brain: Exploring Mechanistic Insights and Therapeutic Potential

**DOI:** 10.3390/ijms262110743

**Published:** 2025-11-05

**Authors:** Osama Sobhi Moaket, Sarah Eyad Obaid, Fawaz Eyad Obaid, Yusuf Abdulkarim Shakeeb, Samir Mohammed Elsharief, Afrin Tania, Radwan Darwish, Alexandra E. Butler, Abu Saleh Md Moin

**Affiliations:** 1School of Medicine, Royal College of Surgeons in Ireland-Bahrain, Busaiteen P.O. Box 15503, Bahrain; 21200278@rcsi-mub.com (O.S.M.); 22200293@rcsi-mub.com (S.E.O.); 21204313@rcsi-mub.com (F.E.O.); 21200513@rcsi-mub.com (Y.A.S.); 21200180@rcsi-mub.com (S.M.E.); afrin126@gmail.com (A.T.); 22200493@rcsi-mub.com (R.D.); 2Research Department, Royal College of Surgeons in Ireland-Bahrain, Busaiteen P.O. Box 15503, Bahrain; abutler@rcsi.com

**Keywords:** GLP-1 receptor agonists (GLP-1RAs), neurodegeneration, Alzheimer’s disease, Parkinson’s disease, stroke, neuroinflammation, synaptic plasticity, clinical trials

## Abstract

Neurodegenerative disorders, including Alzheimer’s disease (AD), Parkinson’s disease (PD), stroke, and depression, are marked by progressive neuronal dysfunction and loss, yet current treatments remain largely symptomatic with limited disease-modifying efficacy. Glucagon-like peptide-1 (GLP-1), an incretin hormone traditionally associated with metabolic regulation, has emerged as a promising neuroprotective agent. Its receptor, GLP-1R, is expressed in key brain regions implicated in cognition, emotion, and motor control, including the hippocampus, frontal cortex, and substantia nigra. GLP-1R agonists (GLP-1RAs) activate multiple intracellular signaling cascades—cAMP/PKA, PI3K/Akt, and MAPK pathways—that collectively promote neuronal survival, enhance synaptic plasticity, reduce oxidative stress, inhibit apoptosis, and modulate neuroinflammation. These agents also regulate autophagy, promote remyelination, and reprogram microglial phenotypes toward anti-inflammatory states. Preclinical models have shown that GLP-1RAs reduce amyloid-β and tau pathology in AD, preserve dopaminergic neurons in PD, protect astrocytes and neural progenitors after ischemic stroke, and alleviate depressive behaviors. Notably, GLP-1RAs such as liraglutide, exenatide, and dulaglutide can cross the blood–brain barrier and have demonstrated safety and potential efficacy in early-phase clinical trials. These studies report attenuation of cortical atrophy, preservation of cerebral glucose metabolism, and improvements in quality of life, though changes in core AD biomarkers remain inconclusive. Ongoing large-scale trials (e.g., EVOKE, ELAD) are further exploring their therapeutic impact. This review consolidates the mechanistic basis and translational potential of GLP-1RAs in age-related neurodegenerative diseases, highlighting both their promise and the challenges that must be addressed in future clinical applications.

## 1. Introduction

Neurodegenerative disorders, such as Alzheimer’s and Parkinson’s disease, are characterized by progressive neuronal loss leading to cognitive decline and motor dysfunction. These conditions represent a significant global health challenge due to their increasing prevalence and the lack of effective disease-modifying therapies. Current treatments primarily offer symptomatic relief without halting disease progression, underscoring the urgent need for novel therapeutic approaches.

Glucagon-like peptide-1 (GLP-1) has emerged as a promising candidate for neuroprotection [1]. Although traditionally recognized for its role in glucose metabolism, recent research has highlighted GLP-1’s significant effects in the central nervous system (CNS), particularly its neuroprotective actions [2]. The GLP-1 receptor (GLP-1R) is widely expressed in brain regions involved in cognition, learning, memory, and motor function, such as the striatum, midbrain, hindbrain, hypothalamus and brainstem in nonhuman primates [3]. Activation of the GLP-1R mediates neuroprotective effects through multiple mechanisms, including reducing neuroinflammation and oxidative stress, modulating synaptic plasticity, improving cognitive function and inhibiting neuronal apoptosis. In AD models, GLP-1 exerts neuroprotective effects by activating the Cyclic Adenosine Monophosphate/Protein Kinase A (cAMP/PKA).

Despite these promising findings, the detailed molecular mechanisms by which GLP-1 exerts its neuroprotective effects remain incompletely understood. Specifically, there is limited knowledge on the long-term effects of GLP-1R activation in neuroprotection and how GLP-1 influences cellular and molecular processes in the brain. This gap in understanding hinders the development of targeted therapies for neurodegenerative diseases. Therefore, the objective of this study is to synthesize current knowledge about GLP-1 and GLP-1R functions in the brain, focusing on their neuroprotective mechanisms and their role in regulating body homeostasis in the context of neurodegeneration. By elucidating the molecular mechanisms underlying GLP-1’s actions in the CNS, we aim to address critical knowledge gaps that could guide the development of new therapeutic strategies leveraging GLP-1’s neuroprotective potential.

## 2. Expression and Distribution of Glp-1 and Glp-1r in Nervous System

### 2.1. The Peripheral Nervous Pathway of Glp-1: Vagus-Driven Central Effects

Understanding the expression and distribution of GLP-1 and its receptor within the CNS is crucial for elucidating their neuroprotective roles and therapeutic potential in neurodegenerative diseases. GLP-1 is synthesized both peripherally and centrally. While enteroendocrine L cells in the gut are the primary peripheral source, GLP-1 is also produced by preproglucagon-expressing neurons located in the nucleus tractus solitarius (NTS) of the brainstem. The central production of GLP-1 allows it to exert direct effects within the brain, bypassing the limitations posed by the blood–brain barrier and rapid degradation of peripheral GLP-1 by dipeptidyl peptidase-4 (DPP-4), which reduces its half-life to less than two minutes [4]. Peripheral GLP-1 influences the brain through two primary pathways: neural and humoral (Figure 1). The neural pathway involves GLP-1 interacting with vagal afferents in the intestinal or hepatic portal area, transmitting signals to the NTS. The humoral pathway, which is a less likely mode of action due to rapid degradation of GLP-1, involves direct entry into the brain via the blood–brain barrier. Consequently, the neural pathway is considered the predominant route for gut-derived GLP-1 to affect central functions [5]. Although GLP-1RAs are frequently described as exerting “central” effects, their pharmacokinetic and BBB profiles differ markedly, shaping the extent of true central exposure. Peptide-based agonists such as exenatide (4.2 kDa) and lixisenatide (4.8 kDa) exhibit measurable but limited BBB permeability, with cerebrospinal fluid (CSF) levels reaching only 0.02% (range: 0.002–0.07%) of plasma concentrations for liraglutide [6], though both liraglutide and lixisenatide demonstrate elevated brain levels 30 min following injection [7]. Liraglutide (3.8 kDa, albumin-bound, t½ ≈ 13 h) shows minimal CSF penetration (6.5 pmol/L CSF vs. 31 nmol/L plasma, representing approximately 0.02%) [6,8], yet a prolonged systemic half-life of 13 h due to albumin binding [9] may sustain indirect neurotrophic and anti-inflammatory signaling. Semaglutide (4.1 kDa, t½ ≈ 7 days [approximately 165–184 h]) likewise demonstrates minimal direct CNS entry but robust peripheral vascular and metabolic modulation, which may confer secondary neuroprotection [10]. Notably, rodents exhibit higher relative CNS uptake than humans due to differences in endothelial transport mechanisms [11], and effective preclinical doses (25–250 nmol/kg) [7] far exceed typical human-equivalent exposures. These interspecies pharmacokinetic disparities must be acknowledged when interpreting “central” mechanisms or extrapolating animal efficacy to clinical settings.

Moreover, GLP-1RAs encompass pharmacologically distinct subclasses rather than a homogeneous entity. Short-acting compounds such as exenatide and lixisenatide yield rapid peak concentrations with transient receptor activation and relatively greater short-term BBB penetration, whereas long-acting analogues, including liraglutide, dulaglutide, and semaglutide, provide sustained systemic exposure but reduced peak brain concentrations due to their albumin binding and extended half-lives [12,13]. Extended-release platforms like PT302, which encapsulates exenatide in biodegradable PLGA microspheres of 20 μm diameter, maintain stable plasma levels from day 10–28 following a single subcutaneous dose and achieve CSF exendin-4 concentrations of 18.3–30 pg/mL compared to <6.9 pg/mL for immediate-release exendin-4 [14] in rodent models. These kinetic distinctions likely underlie differences in preclinical efficacy profiles, where short-acting agents often outperform in acute models, correlating with their superior BBB penetration, while long-acting molecules exert more consistent anti-inflammatory and metabolic benefits in chronic paradigms [10]. Recognizing formulation-specific dynamics is essential for accurate interpretation of mechanistic and translational findings.

### 2.2. Glp-1 Receptors in the Central Nervous System (CNS): Regional Density and Physiological Roles

Unlike the limited areas that produce GLP-1, GLP-1R is widely distributed throughout the CNS, with expression patterns that correlate with key neurophysiological functions. The receptor’s presence in specific brain regions underscores its potential impact on neuroprotection, cognition, appetite regulation and emotional processing. GLP-1R-expressing cells were observed throughout the rostro-caudal extent of the mouse brain [15]. The largest numbers of GLP-1R cells were observed in regions associated with autonomic and behavioural control of energy balance, such as the paraventricular nucleus (PVN), arcuate nucleus (ARC), dorsomedial hypothalamic nucleus (DMH) and central amygdala. A schematic representation of the distribution of GLP-1R neurons throughout the mouse brain is given in Figure 2. The hypothalamus plays a central role in regulating energy balance, appetite, and glucose homeostasis. GLP-1R expression is notably high in hypothalamic nuclei such as the arcuate nucleus, paraventricular nucleus and dorsomedial nucleus [16].

### 2.3. Species Differences of Glp-1r Expression in Brain: Implications for Translation

Comparative studies have highlighted both similarities and differences in GLP-1R distribution across species, which have significant implications for translational research. In rodents, GLP-1R expression is prominent in the hypothalamus, aligning with the strong influence of GLP-1 on feeding behaviour in these animals. Non-human primates also show substantial hypothalamic expression but with notable differences in cortical regions. Based on transcriptomic mapping from the Human Protein Atlas (Brain Atlas), GLP-1R mRNA is detected across multiple brain regions including the cerebral cortex, hippocampal formation, amygdala, basal ganglia, thalamus, cerebellum, and white matter at normalized transcripts per million (nTPM) ≥ 1.0. The Human Protein Atlas also indicates that the highest GLP-1R mRNA levels occur in the hypothalamus (nTPM~8.0), followed by the pons and medulla oblongata (nTPM~3.0) [17]. However, recent mapping of the human brain revealed that, while GLP-1R is present in the hippocampus across species, humans exhibit the highest expression in the frontal cortex rather than the hypothalamus [18] (Figure 3). Additionally, GLP-1Rs are absent in the human cerebellum, contrasting with their presence in the cerebellum in animal models. Humans also display GLP-1R expression throughout the cerebral cortex, except for the orbitofrontal cortex, whereas animal models show limited cortical expression. These species-specific differences underscore the importance of cautious interpretation when extrapolating animal data to humans. They highlight the need for human-specific studies to accurately assess the therapeutic potential of GLP-1R agonists in neurodegenerative diseases [18]. Some data indicate that GLP-1R expression in the mouse brain is more regionally concentrated, for example, the mouse brain exhibits region-enriched GLP-1R expression in the hindbrain nuclei and hypothalamus [15,16]. In contrast, the human brain displays a more uniform GLP-1R distribution with low regional specificity and no single dominant site (except the hypothalamus as discussed above, which may draw some similarities to mice) [17]. GLP-1R in mice was observed in regions associated with autonomic and behavioural control of energy balance, such as the paraventricular nucleus (PVN), arcuate nucleus (ARC), dorsomedial hypothalamic nucleus (DMH) (all parts of the hypothalamus) and central amygdala [16].

These interspecies discrepancies extend beyond regional expression to functional receptor pharmacology. For example, rodents display high GLP-1R expression in thyroid C-cells and alveolar tissues, sites with minimal or absent expression in primates and humans [19,20], highlighting why rodent toxicology findings such as C-cell hyperplasia do not translate clinically, as humans and cynomolgus monkeys had low GLP-1 receptor expression in thyroid C-cells, and GLP-1 receptor agonists did not activate adenylate cyclase or generate calcitonin release in primates [19]. Moreover, receptor coupling efficiency and downstream signaling bias differ subtly between species; while specific data on Gαs-biased signaling strength differences between murine and human GLP-1Rs remains limited in the literature, β-arrestin recruitment and receptor desensitization pathways play critical roles in GLP-1R function, with reduced β-arrestin recruitment associated with prolonged cAMP signaling and enhanced metabolic efficacy [21,22], and compounds that reduce β-arrestin recruitment and retain GLP-1R at the plasma membrane produce greater long-term insulin release with faster agonist dissociation rates [23]. Differences in blood–brain barrier permeability, receptor turnover, and neural network integration further limit direct extrapolation from animal data. Consequently, translational studies should prioritize humanized or non-human primate models and early-phase trials employing PET-based receptor occupancy mapping to verify central engagement of GLP-1RAs in humans. While 68Ga-NODAGA-exendin-4 PET showed significant uptake in the pituitary area of obese subjects, no significant uptake was found in other parts of the brain [24], and PET imaging using GLP-1R binding peptides has been employed for monitoring biodistribution and receptor occupancy in various tissues including the pancreas [25].

## 3. Effect of Glp-1r Expression in Different Regions of Brain Functionality

### 3.1. Glp-1r in the Hypothalamus: Appetite and Energy Balance

Activation of GLP-1Rs in these areas suppresses appetite and enhances satiety, contributing to metabolic homeostasis. This high expression suggests that GLP-1Rs in the hypothalamus are a critical mediator of feeding behaviour and energy expenditure [26]. Activation of GLP-1Rs in these areas suppresses appetite and enhances satiety, contributing to metabolic homeostasis. The relatively high expression of GLP-1Rs in the hypothalamus supports their critical role in controlling feeding behaviour. In human studies, using exenatide (GLP-1R agonist) in individuals with type 2 diabetes and obesity has been shown to decrease food intake, underscoring the receptor’s importance in appetite regulation [27].

### 3.2. Glp-1r in the Hippocampus and Cerebral Cortex: Learning, Memory, and Executive Control

GLP-1Rs are expressed in the hippocampus and various regions of the cerebral cortex, including the frontal, prefrontal and parietal cortices. In the hippocampus, GLP-1R activation enhances synaptic plasticity, learning and memory formation [28]. The frontal cortex, associated with executive functions and decision-making, exhibits the highest levels of GLP-1R expression in humans [18]. GLP-1R activation in the hippocampus enhances synaptic plasticity, learning and memory formation [29]. In the cerebral cortex, GLP-1Rs are expressed in various regions, including the frontal, prefrontal and parietal cortices (Figure 2). The frontal cortex, associated with executive functions and decision-making, exhibits the highest levels of GLP-1R expression in cortical areas of the brain according to some reports [18], but stronger and more comprehensive analyses do not support a unique “highest” region in humans [17]. These findings indicate that GLP-1R signalling in these regions may play a significant role in cognitive processes and could be targeted for treating cognitive deficits in neurodegenerative diseases.

### 3.3. Glp-1r in Other Brain Regions: Mood, Reward, and Pain/Stress Modulation

Additional regions expressing GLP-1Rs include the amygdala, the bed nucleus of the stria terminalis, ventral midbrain, periaqueductal grey matter and the lateral septum. In these areas, GLP-1Rs interact with neurotransmitter systems such as dopamine, serotonin and glutamate pathways, influencing mood regulation and reward-related behaviours. For example, GLP-1R activation in the lateral septum modulates dopamine release, affecting reward processing and potentially impacting addictive behaviours [30]. GLP-1R has been detected in nucleus accumbens, ventral tegmental area, amygdala, and prefrontal cortex. In these areas, GLP-1Rs interact with neurotransmitter systems such as dopamine, serotonin and glutamate pathways, influencing mood and emotion regulation, reward-related behaviours and stress responses [31]. In a recent randomized controlled trial, Hendershot et al. (2025) demonstrated that low-dose Semaglutide significantly reduced alcohol consumption and craving in adults with alcohol use disorder [32]. In addition, GLP 1 receptor agonists (GLP 1RAs) influence the dopamine transporter (DAT) function and dopamine release within mesolimbic pathways, which might contribute to behaviours against addiction [33]. These findings have sparked interest in potential neuropsychiatric effects of GLP 1 signalling such as its possible use in alleviating symptoms of depression or anxiety or improving cognitive deficits especially in neurodegenerative disorders. However, these ideas remain largely speculative. Evidence for mood-enhancing or anxiolytic effects of GLP 1RAs in humans is limited and inconclusive. Carefully designed in vivo studies and robust clinical trials are essential to confirm any potential psychiatric benefits.

### 3.4. Spatial Mapping of Glp-1r and Its Translational Value for Neurodegenerative Disorders

The regional distribution of GLP-1R in the human brain aligns with areas affected by neurodegenerative diseases. The high expression in the hippocampus and frontal cortex suggests that GLP-1R agonists could have therapeutic effects on cognitive functions impaired in conditions like Alzheimer’s disease (AD) [28]. GLP-1R activation may enhance neurogenesis, promote synaptic plasticity and inhibit apoptotic pathways, contributing to neuroprotection [34]. Furthermore, the presence of GLP-1Rs in regions involved in mood and reward processing indicates potential benefits for neuropsychiatric symptoms associated with neurodegenerative diseases. By influencing neurotransmitter systems, GLP-1R agonists might ameliorate depression, anxiety and motivational deficits commonly observed in these conditions [35]. While the widespread distribution of GLP-1Rs offers promising therapeutic avenues, several challenges persist. The differential expression patterns between humans and animal models (Table 1) raise questions about the translatability of preclinical findings. For instance, the lower hypothalamic expression of GLP-1Rs in humans compared to rodents suggests that appetite suppression observed in animal studies may not directly correlate with human outcomes [35]. Understanding why certain brain regions exhibit higher GLP-1R expression is essential for targeted therapy development. Factors such as regional differences in receptor regulation, neuronal subtype specificity and varying roles of GLP-1R signalling pathways could contribute to these patterns. Elucidating these mechanisms may enhance the effectiveness of GLP-1R-based interventions. Limitations in current data include a lack of longitudinal studies assessing the long-term effects of GLP-1R activation in humans and insufficient exploration of how GLP-1R’s interactions with other cellular pathways influence neuroprotection. Addressing these gaps through advanced imaging techniques, molecular studies and clinical trials will be critical for translating basic research into effective treatments.

## 4. Molecular Mechanisms of Glp-1 Action in Neurons

Understanding the molecular mechanisms by which GLP-1 exerts neuroprotective effects is crucial for developing effective therapies for neurodegenerative diseases. GLP-1 and its receptor influence various intracellular signaling pathways that collectively promote neuronal survival, enhance myelination, prevent demyelination, modulate autophagy and apoptosis, interact with microglia, and reduce oxidative stress. This section explores these interconnected mechanisms, highlighting their implications for neuroprotection and potential therapeutic applications.

### 4.1. Intracellular Signaling Mechanisms of Glp-1 in Neurons

Upon GLP-1 binding, the GLP-1R-coupled Gα subunit is activated by GTP, which then stimulates adenylate cyclase, converting adenosine triphosphate (ATP) into cyclic adenosine monophosphate (cAMP). Increased cAMP levels activate protein kinase A (PKA) through phosphorylation. PKA activation promotes cell growth and survival by enhancing neurotransmitter release and synaptic plasticity [37]. Activation of GLP-1R has been shown to protect cortical neurons from oxidative DNA damage by stimulating cyclic AMP response element-binding protein (CREB), which induces apurinic/apyrimidinic endonuclease 1 (APE1) expression for DNA repair via the base excision repair (BER) pathway. APE1 expression was found to be reduced by phosphatidylinositol-3 kinase (PI3K) inhibition but not by mitogen-activated protein kinase (MEK) suppression, and administration of exendin-4 (a GLP-1 analogue) was observed to enhance DNA repair in ischemic stroke rat brains [38]. Insulin rapidly induces tyrosine phosphorylation and catalytic activity of PI3K. Evidence suggests that PKB/Akt is of major importance in mediating the effects of PI3K in neuronal survival. In vivo, activated PKB/Akt in the hippocampus is associated with neuronal protection against hypoxic stress and nitric oxide toxicity [39]. PI3K activates protein kinase B (Akt) via phosphorylation, supporting synapse formation and protection. Akt also increases the production of myelin proteins, such as myelin protein zero (MPZ) and peripheral myelin protein 22 (PMP22), contributing to remyelination and Schwann cell health [40] (Figure 4).

### 4.2. Glp-1 Enhances Myelination

Exendin-4 (Ex-4), a GLP-1RA, has demonstrated neurotrophic capabilities by enhancing the survival and proliferation of Schwann cells and promoting neurite outgrowth from dorsal root ganglion neurons. In studies using immortalized adult rat Fischer Schwann cells (IFRS1), Ex-4 enhanced cell migration towards neurites and upregulated the expression of MPZ and PMP22, essential components of the myelin sheath, and the effects were mediated through the activation of the PI3K/Akt pathway [41].

### 4.3. Glp-1 Prevents Demyelination by Inhibiting Neuronal Inflammation

Neuroinflammation plays a pivotal role in the progression of demyelinating diseases. GLP-1R activation exerts anti-inflammatory effects that can mitigate neuronal damage. Administration of Ex-4 in animal models reduced demyelination, microglial activation and clinical symptoms of multiple sclerosis (MS) [42]. Ex-4 suppressed mRNA expression of pro-inflammatory cytokines such as interleukin-1β (IL-1β), IL-6, IL-17, and tumor necrosis factor-alpha (TNF-α), which are implicated in excessive innate immune responses and MS progression [42]. Similarly, Dulaglutide, another GLP-1RA, improved clinical symptoms by reducing lymphocyte infiltration and demyelination when administered subcutaneously. It also suppressed encephalitogenic Th1/Th17 cells in the CNS, further attenuating the inflammatory response [43]. These anti-inflammatory actions contribute to preserving myelin integrity and neuronal function. By modulating immune responses, GLP-1R activation offers a multifaceted approach to preventing demyelination and promoting neural health.

### 4.4. Glp-1 Potentiates Axonal Regeneration Through Modulating Autophagy and Neuronal Apoptosis

Ex-4 administration in spinal cord injury (SCI) models promoted autophagy and suppressed apoptosis, facilitating neuronal survival and regeneration. Higher Ex-4 doses showed stronger neuroprotective effects, evidenced by increased LC3-II and Beclin 1 expression and decreased caspase-3 levels, key markers of autophagic activity [44]. Additionally, a lack of Beclin 1 in cultured neurons and transgenic mice leads to the deposition of Aβ peptides, while its overexpression helps to reduce their accumulation. This highlights the positive role of autophagy in AD by lowering the levels of Aβ, which are a defining feature of the disease’s pathology [45]. In neurons, under stress conditions or specific stimuli, Akt signaling can promote autophagy by stimulating the production of phosphatidylinositol-3-phosphate (PtdIns3P), which aids autophagosome formation by recruiting autophagy-related proteins. Beclin 1, as part of the Class III PI3K complex, contributes to PtdIns3P production, further supporting autophagosome formation [46] (Figure 4). Concurrently, Ex-4 reduced apoptosis by decreasing caspase-3 expression, a crucial executor of apoptosis. By balancing autophagy and apoptosis, GLP-1R activation supports the clearance of damaged cellular components while preventing unnecessary cell death.

### 4.5. Glp-1 Action in Microglia

Microglia are the primary immune cells of the CNS, essential for maintaining homeostasis and responding to injury or pathological stimuli. While transient microglial activation supports tissue repair, chronic activation drives sustained neuroinflammation, contributing to neurodegenerative processes. GLP-1RAs have emerged as powerful modulators of microglial behaviour, shifting them from pro-inflammatory to anti-inflammatory phenotypes. In models of AD, GLP-1RAs such as liraglutide have been shown to significantly reduce activated microglia numbers in the hippocampus and cortex by approximately 50%, accompanied by decreased pro-inflammatory cytokine levels like IL-1β and TNF-α [47]. Furthermore, dual GLP-1/GIP receptor agonists exert superior anti-inflammatory effects by dampening microglial activation and reducing neurotoxic mediators [48]. In addition to attenuating neuroinflammation, GLP-1RAs enhance microglial phagocytic capacity, promoting clearance of amyloid-β and other pathological aggregates, thereby contributing to neuroprotection [49]. These findings underscore the potential of GLP-1RAs as promising therapeutics in neurodegenerative conditions such as AD and PD, where microglia-driven inflammation plays a pivotal pathogenic role [1].

### 4.6. Glp-1 Enhances Neuronal Survival

GLP-1 receptor activation has emerged as a powerful neuroprotective mechanism by attenuating oxidative stress and promoting neuronal resilience. Under diabetic and neurodegenerative conditions, excessive reactive oxygen species (ROS) production—partly due to elevated levels of advanced glycation end products (AGEs)—leads to oxidative damage, mitochondrial dysfunction, and neuronal death. GLP-1RAs activate the PI3K/Akt pathway, which in turn stimulates the phosphorylation of the cAMP response element-binding protein (CREB). CREB upregulates the expression of antioxidant enzymes and survival factors, reducing oxidative stress and improving neuronal function [50]. Furthermore, GLP-1RAs downregulate pro-apoptotic proteins such as Bax and cleaved caspase-3 while enhancing anti-apoptotic proteins like Bcl-2 through the combined action of cAMP-PKA, PI3K/Akt, and CREB signaling pathways (as indicated by broken lines in Figure 4), thereby promoting neuronal survival and protecting against apoptosis [51]. This integrated signaling cascade not only protects neurons from apoptosis but also facilitates neuronal differentiation and neurite outgrowth, further contributing to neuronal repair and plasticity [52]. These effects make GLP-1RAs promising therapeutic candidates for neurological diseases characterized by oxidative stress and neuronal apoptosis, such as Alzheimer’s and Parkinson’s diseases.

## 5. Neuroprotective Actions of Glp-1 (Summarized in Table 2)

### 5.1. Glp-1 in Alzheimer’s Disease (AD)

AD is the most common neurodegenerative disease, pathologically characterized by amyloid-β (Aβ) plaque deposition and neurofibrillary tangles. A study investigating the relationship between GLP-1 and AD reported that decreased plasma GLP-1 levels were observed in older individuals, and even lower levels were found in patients with AD [53]. In addition, elevated plasma GLP-1 levels were linked to improved cognitive function, as assessed by Mini-Mental State Examination scores, while demonstrating an inverse correlation with plasma pTau181 levels, a biomarker associated with the progression of AD [53]. Mechanistically, GLP-1 has been shown to enhance mitochondrial activity and promote the cleavage of amyloid precursor protein (APP), reducing the production of Aβ [53,54]. Ex-4 attenuates neuroinflammation by shifting microglia from a pro-inflammatory M1 phenotype to an anti-inflammatory M2 state in both in vitro and in vivo settings via restoration of Arf and Rho GAP adapter protein 3 (ARAP3) expression; collectively, this points to the PI3K/ARAP3/RhoA signaling axis as pivotal for Ex-4’s anti-inflammatory effects [55]. In an Alzheimer’s disease mouse model, liraglutide suppresses microglial activation of the NLR family pyrin domain containing 3 (NLRP3) inflammasome, thereby reducing pro-inflammatory cytokine production and Aβ plaque burden [56]. Semaglutide, a GLP-1 receptor agonist, diminishes seizure severity and cognitive deficits by inhibiting the NLRP3 inflammasome pathway triggered by lipopolysaccharide (LPS) and nigericin, while also lowering lactate dehydrogenase (LDH) release from microglia [57].

#### 5.1.1. Efforts in Elucidating the Role of GLP-1RAs in Dementia Management

GLP-1 receptor agonists (GLP-1RAs) engage brain insulin-signaling pathways, cross the blood–brain barrier, and exert anti-inflammatory and mitochondrial-stabilizing effects relevant to neurodegeneration [58]. In APP/PS1 Alzheimer’s model mice, liraglutide improved object recognition and maze performance and enhanced neural plasticity, with no benefit in normal controls [59]. Peripheral liraglutide (0.2 mg/kg daily for 28 days) in female AD mice reduced cortical Aβ1-42 without consistent effects on other assessed domains [60]. Exenatide improved short- and long-term memory in an amyloid-independent mouse model [61]. In a transgenic AD model, prolonged exenatide administration prevented cognitive decline, reduced Aβ1-42 deposition, and protected synapses and mitochondria [62]. In a randomized, placebo-controlled trial in moderate Parkinson’s disease, weekly exenatide produced motor benefits that persisted after treatment cessation, supporting exploration in Lewy body disorders [63]. Observational population data suggest GLP-1RA exposure may be associated with lower risks of several dementias, supporting a possible class effect that extends beyond metabolic control [64]. A large target-trial emulation in older adults with type 2 diabetes found no clear overall difference in incident dementia between GLP-1RAs and DPP-4 inhibitors, underscoring the need for randomized dementia-focused trials [65]. A contemporary meta-analysis of randomized trials indicates that among cardioprotective glucose-lowering agents, GLP-1RAs show a signal toward reduced all-cause dementia, while emphasizing heterogeneity and the necessity of dedicated cognitive endpoints [65]. Collectively, preclinical and early clinical evidence supports GLP-1RAs as promising candidates for disease-modifying therapy across dementia spectra—including Lewy body dementia—while highlighting the priority for adequately powered, dementia-specific randomized trials that include DLB cohorts and cognitive/functional outcomes. Despite promising preclinical and biomarker data, a systematic review reported that, in the analysed studies, GLP-1 receptor agonist (GLP-1 RA) therapy did not demonstrate significant differences in Aβ and tau biomarkers compared to placebo groups. Likewise, there were no notable improvements in cognitive outcomes between the treated and placebo groups. However, GLP-1 RA treatment did show metabolic benefits, including reduced body mass index and improved glucose levels, and there were indications of potential neuroprotective effects on cerebral glucose metabolism and blood-brain glucose transport capacity [66]. In contrast, a phase II trial of liraglutide in AD patients demonstrated preserved cortical and temporal lobe volume and modest cognitive improvements [67].

#### 5.1.2. Diabetes and Dementia: Connecting Brain Insulin Signaling, GLP-1 Pathways

Previous studies illustrated the relationship between diabetes and cognitive decline through a sequence of events that describe the progression of cognitive impairment, ultimately leading to dementia, including Alzheimer’s disease [68]. The gradual decline in diabetic patients begins with diabetes-associated cognitive decrements, progressing to mild cognitive impairment, and eventually leading to dementia. Therefore, AD is classified as type 3 diabetes mellitus due to the potential link between insulin resistance and AD’s pathogenesis [69]. Considering that insulin is a neuroprotective factor that affects brain function and Aβ deposition, and that a disruption in insulin signalling pathways or insulin resistance would lead to excessive amounts of reactive oxygen species, inflammatory cytokines, dysregulation of tau protein metabolism and crucial enzymes such as beta-secretase 1 (BACE-1) and insulin-degrading enzyme (IDE), ultimately leading to brain insulin resistance and neuronal loss, this underscores the potential benefits of antidiabetic drugs, especially GLP-1 RAs, in preventing cognitive decline and delaying the onset of AD [54]. Moreover, Insulin-like growth factor-1 (IGF-1) and GLP-1 play important roles in brain development and regulating glucose levels, with GLP-1 sharing similarities with IGF-1 in providing brain protection. These proteins influence cellular pathways to combat neuronal death in neurodegenerative diseases, with their ability to cross the blood–brain barrier and support functions like synaptic formation and neuronal plasticity. Their impact on the nervous system positions them as promising targets for treating various brain disorders [70]. Another study focusing on type 2 diabetes (T2D) discovered a connection between the insulin signaling cascade and AD. Impairment of the insulin signaling pathway causes the tau protein to become hyperphosphorylated, which is essential for AD pathogenesis. Insulin has a positive impact on cognition, with the intranasal administration route showing potential benefits for improved cognition. Conversely, however, insulin administered intravenously and subcutaneously may raise the risk of dementia [71]. In addition, patients suffering from either type 1 diabetes (T1D) or T2D may experience hypoglycemic episodes, which may precipitate both vascular and non-vascular dementia. This is because neuroglycopenia following a hypoglycaemic episode may result in neuronal death, cognitive impairment, and dementia [72].

### 5.2. Glp-1 in Parkinson’s Disease (PD)

PD is a progressive neurodegenerative disorder that causes both motor and non-motor symptoms. Numerous studies have investigated the neuroprotective effects of the newly emerging GLP-1 agonists. A systematic review and meta-analysis illustrated that treatment with GLP-1 agonists in preclinical rodent models of PD showed a substantial reduction in motor symptoms and dopaminergic neurotransmission, suggesting neuroprotective benefits in these models [73]. Another study examined the neuroprotective effects of the GLP-1R agonist liraglutide in an Mitochondrial permeability transition pore (MPTP)-induced PD model, where MPTP exposure led to mitochondrial dysfunction, impaired autophagy, and increased cell apoptosis [74]. Liraglutide counteracted these effects by activating proliferator-activated receptor-γ coactivator 1α (PGC-1α), improving mitochondrial quality control, and reducing neurotoxicity. However, downregulating PGC-1α eliminated the neuroprotective effects of liraglutide, suggesting that PGC-1α plays a key role in regulating mitochondrial biogenesis, function, autophagy, and apoptosis [74]. Furthermore, a study investigated the relationship between insulin-resistant neurons and PD on iPSC (induced pluripotent stem cells) models of synucleinopathy, shedding light on a potential pathway that can be targeted to treat PD [75]. The neurons of the iPSC models showed abnormal insulin response and signalling, which resulted in the blockage of neuroprotective pathways and the activation of stress pathways. As a result, α-synuclein buildup, compromised protein synthesis and neuronal death was observed. However, Exenatide, a GLP-1 RA, reversed the effects of insulin resistant neurons by activating the neuroprotective Protein Kinase B (Akt) signalling, inhibiting MAPK (mitogen-activated protein kinase) pathways involved in stress response and enhancing the activity of mitochondria and lysosomes, in addition to decreasing oxidative stress, α-synuclein aggregates, and inflammatory cytokine IL-6 (interleukin 6) [75]. Another study mentioned that, despite the morphological evidence that identifies the presence of GLP-1 receptors in the substantia nigra pars compacta (SNc), the effect of GLP-1 on the firing patterns of dopamine-producing neurons remained unclear until extracellular in vivo recordings were conducted to show that GLP-1 is involved in controlling the firing activity of neurons. This was evident through the reduced firing rate of dopaminergic neurons located in the SNc and GLP-1’s ability to stimulate channels involved in the excitation of neurons [76]. Another study reported that GLP-1 agonists played a significant role in improving motor symptoms associated with PD in phase II clinical trials [67]. In another clinical trial that involved PD patients, Liraglutide treatment resulted in notable enhancements in patients’ overall quality of life, daily activities and non-motor symptoms. This was evident through the non-motor symptom scale scores, PD Questionnaire results and Parkinson’s Anxiety Scale scores. However, there was no change in cognitive performance during the duration of the clinical trial [77]. Another study demonstrated a clear association between abnormal gut–brain communication and neurodegenerative diseases; PD, in particular, is characterised by an imbalance in gut microbiota and neuroinflammation, further supporting the role of GLP-1 in neuroprotection [78].

### 5.3. Glp-1 in Huntington’s Disease

Huntington’s disease (HD) is a progressive and fatal neurodegenerative disease caused by CAG repeat expansion in the coding region of huntingtin (HTT) protein. Huntington’s disease features mutant HTT (mHTT)-driven autophagy defects and oxidative stress, and epidemiology shows higher type-2 diabetes prevalence in HD; in neurons, mHTT overexpression impaired insulin signaling and triggered apoptosis. leading to progressive motor, cognitive, and psychiatric symptoms [79]. Liraglutide (GLP-1 analogue) restored insulin sensitivity, improved cell viability, upregulated antioxidant pathway genes to reduce oxidative stress, and activated AMPK-dependent autophagy to lessen HTT aggregate accumulation—suggesting GLP-1 may mitigate mHTT neurotoxicity [80]. In a 3-nitropropionic acid (3-NP) rat model of HD, liraglutide improved motor function. It boosted protective brain pathways (miR-130a, BDNF/TrkB, PI3K–Akt–CREB, β-catenin), while reducing stress and cell death signals (sortilin, p75NTR, oxidative damage, Bax, caspase-3), suggesting strong antioxidant, anti-apoptotic, and neurotrophic effects [81]. Exendin-4 improved peripheral glucose regulation reduced cellular pathology in brain and pancreas and was associated with fewer mHTT aggregates in islet and brain cells. These effects translated into better motor performance and significantly longer survival in the Huntington’s disease mouse model [82]. These findings align with broader reviews showing that incretin hormones, including GLP-1, can cross the blood–brain barrier, regulate mitochondrial function, and mitigate oxidative stress, reinforcing their promise as therapeutic agents in HD and other neurodegenerative disorders.

**Table 2 ijms-26-10743-t002:** GLP-1R therapeutic benefits and mechanisms in neurodegenerative diseases.

Neurodegenerative Diseases	Therapeutic Benefits of Using GLP-1Rs	Mechanism of Action	References
Alzheimer’s diseases	Better cognition associated with higher plasma GLP-1 levels (MMSE). Reduced Aβ burden and improved learning/memory in multiple AD models. Preserved cortical/temporal lobe volume with modest cognitive benefit in a phase II trial.	Enhances mitochondrial function and promotes APP cleavage → decreased Aβ. Shifts microglia M1 → M2 via ARAP3; PI3K/ARAP3/RhoA axis involvement. Inhibits microglial NLRP3 inflammasome; lowers pro-inflammatory cytokines. Semaglutide suppresses LPS/nigericin-induced NLRP3 activity; lowers microglial LDH.	[53,55,56,57,59,60,61,62,67]
Dementia	CNS engagement relevant to neurodegeneration (class rationale). Observational data associate GLP-1RA exposure with lower risks of several dementias. Target-trial emulation shows no clear overall difference vs. DPP-4 inhibitors; meta-analysis signals possible dementia reduction among cardioprotective agents. Systematic review: metabolic benefits; mixed/negative cognitive and biomarker changes.	Crosses the blood–brain barrier; engages brain insulin signaling; anti-inflammatory and mitochondrial-stabilizing effects.	[58,64,65,66]
Parkinson’s disease	Reduces motor symptoms and supports dopaminergic function in preclinical models. Exenatide provides motor benefits that persist post-treatment; clinical trials show improved non-motor symptoms and quality of life with liraglutide. Links to gut–brain axis dysbiosis and neuroinflammation support GLP-1 neuroprotection.	Liraglutide activates PGC-1α → improved mitochondrial quality control, autophagy, and reduced apoptosis. Exenatide activates Akt, inhibits MAPK stress pathways; decreases α-syn aggregates, IL-6; increases mitochondrial/lysosomal activity in iPSC models. GLP-1 modulates substantia nigra dopaminergic neuron firing and excitatory channel activity.	[67,73,74,75,76,77,78]
Huntington’s disease	Restores insulin sensitivity; improves cell viability; reduces mHTT aggregates via autophagy activation. Improves motor function and survival; ameliorates peripheral glucose dysregulation; less cellular pathology in brain and pancreas.	Activates AMPK-dependent autophagy; upregulates antioxidant genes; mitigates oxidative stress. ↑ miR-130a, BDNF/TrkB, PI3K-Akt-CREB, β-catenin; ↓ sortilin, p75NTR, oxidative damage, Bax, caspase-3.	[79,80,81,82]

AD, Alzheimer’s disease; PD, Parkinson’s disease; LBD, Lewy body dementia; GLP-1, glucagon-like peptide-1; GLP-1RA, GLP-1 receptor agonist; APP, amyloid precursor protein; Aβ, amyloid-β; MMSE, Mini-Mental State Examination; ARAP3, Arf and Rho GAP adapter protein 3; PI3K, phosphoinositide 3-kinase; RhoA, Ras homolog family member A; NLRP3, NOD-like receptor family pyrin domain-containing 3; LPS, lipopolysaccharide; LDH, lactate dehydrogenase; BBB, blood–brain barrier; CNS, central nervous system; DPP-4, dipeptidyl peptidase-4; PGC-1α, peroxisome proliferator-activated receptor-γ coactivator-1α; Akt, protein kinase B; MAPK, mitogen-activated protein kinase; IL-6, interleukin-6; iPSC, induced pluripotent stem cell; α-syn, alpha-synuclein; SNc, substantia nigra pars compacta; AMPK, AMP-activated protein kinase; BDNF, brain-derived neurotrophic factor; TrkB, tropomyosin receptor kinase B; CREB, cAMP response element-binding protein; mHTT, mutant huntingtin; HTT, huntingtin; IDE, insulin-degrading enzyme; BACE-1, beta-secretase 1; IGF-1, insulin-like growth factor-1; PDQ, Parkinson’s Disease Questionnaire; NMSS, Non-Motor Symptoms Scale. ↑ indicates increase and ↓ indicates decrease of levels.

## 6. Therapeutic Strategies for Neurodegenerative Disorders

Recent studies highlight that GLP-1 and GLP-1RAs offer promising therapeutic potential in the treatment of neurodegenerative diseases. Their neuroprotective actions include reducing oxidative stress, inhibiting neuronal apoptosis, and improving mitochondrial function—mechanisms essential for counteracting the chronic neuroinflammation and neuronal loss observed in disorders such as Alzheimer’s and Parkinson’s disease. Furthermore, the established link between chronic neuroinflammation, insulin resistance, and type 2 diabetes (T2D) suggests that therapies targeting T2D with incretin-based agents may also alleviate neuroinflammatory processes, offering a dual benefit for metabolic and neurological health [1].

### 6.1. Glp-1 in Neuroprotection: Evidence from Animal Models

Substantial preclinical evidence supports the neuroprotective potential of GLP-1RAs in models of neurodegenerative disorders. In cultured SH-SY5Y human neuroblastoma cells, liraglutide pretreatment significantly reduced oxidative stress and promoted neuronal survival and differentiation, largely through activation of the PI3K-Akt or Akt-STAT3 signaling pathway [83,84].

#### 6.1.1. Studies in Mice

Liraglutide, tested extensively in SH-SY5Y cells [85] and diverse mouse models (APP/PS1, APP/PS1xdb/db, 5xFAD, triple transgenic APP/PS1/Tau) [86,87,88,89] showed consistent reduction in tau hyperphosphorylation, neuroinflammation, and amyloid load. It enhanced autophagy, prevented synapse loss, improved insulin signaling, and restored neurovascular integrity. Across models, administration methods included systemic injection, intranasal delivery, and cell-based treatments, with overall responses showing improved cognitive outcomes, reduced amyloid pathology, and protection against neuroinflammatory and oxidative stress mechanisms in AD. Another GLP-1RA, Dulaglutide, administered in icv-STZ-induced AD mouse models, improved learning and memory via the PI3K/Akt/GSK3β pathway and efficiently crossed the blood–brain barrier [90].

Exenatide treatment in 3xTg-AD mice with streptozotocin (STZ)-induced diabetes not only improved glucose metabolism by elevating plasma insulin and reducing plasma glucose and HbA1c levels but also reduced brain Aβ levels, highlighting its potential benefit in managing AD associated with T2D [91].

Exenatide has demonstrated similar effects in preclinical studies. In 5xFAD transgenic mice, exenatide treatment improved the cognitive performance in the Morris water maze and reduced neuroinflammation and oxidative stress in astrocytes, potentially via the inhibition of the NLRP2 inflammasome [92]. In PD models, for example, in MitoPark mice, treatment with a sustained-release formulation of exenatide (PT320) showed neuroprotective benefits, such as enhanced motor function and improved dopamine signaling in midbrain networks [93].

In the Tg2576 AD mouse model, an eight-month course of intranasal administration of exenatide combined with insulin resulted in enhanced learning abilities and a reduction in cortical amyloid-beta levels, although the decrease in amyloid-beta was not statistically significant [94]. Additionally, prolonged treatment with exenatide improved both short-term and long-term memory in PS1-KI mice [61], likely attributed to elevated lactate dehydrogenase activity and enhanced anaerobic glycolysis.

Beyond these neuroprotective effects, exendin-4 has also been found to exert notable effects on synaptic plasticity, calcium regulation, and cell survival pathways. In icv-Aβ mice, exendin-4 increased the phosphorylation of CREB and elevated BDNF levels, contributing to enhanced synaptic plasticity via increased expression of membrane GluR1 subunits and postsynaptic density protein-95 (PSD-95). These outcomes were associated with upregulated α-secretase activity through ADAM10 membrane trafficking [95].

Lixisenatide, another GLP-1R agonist that can cross the blood–brain barrier, has shown neuroprotective properties, although research is still limited. In an APP/PS1/tau mouse model of Alzheimer’s disease, lixisenatide reduced tau neurofibrillary tangles, amyloid-β plaques and neuroinflammation, as indicated by decreased microglial activation in the hippocampus [96]. These effects were linked to enhanced PKA/CREB pathway signaling and inhibition of the p38/MAPK pathway as a result of GLP-1R activation [96]. Furthermore, lixisenatide treatment improved motor function in a PD MPTP mouse model [97].

#### 6.1.2. Studies in Rats

Liraglutide also demonstrated memory improvements in icv-STZ-induced rat models and non-human primates infused with Aβ oligomers. In different AD rat models, exendin-4 has demonstrated a variety of neuroprotective properties. In rats, exendin-4 administration countered Aβ1-42-induced impairments in the hippocampal CA1 region by restoring long-term potentiation (LTP), cAMP, and phosphorylated CREB levels, while also regulating calcium homeostasis through modulation of intracellular calcium concentrations, highlighting its neuroprotective role [98]. In the icv-STZ rat model, exendin-4 prevented memory deficits and neuronal apoptosis within the hippocampus, promoted cell proliferation, and stimulated synaptogenesis [99]. Likewise, in an icv-Aβ-injected AD rat model, exendin-4 improved memory function, lowered Aβ levels, and restored both acetylcholine levels and mitochondrial function, potentially through the PI3K/Akt signaling pathway [100]. In another icv-STZ rat model, exendin-4 was shown to reduce brain TNF-α levels, maintain choline acetyltransferase (ChAT) activity, enhance cognitive performance, and protect hippocampal neurons from loss [101]. Furthermore, in vitro studies revealed that exendin-4 alleviated neuronal damage induced by high glucose and oxidative stress, and in the icv-STZ-induced rat model, it improved learning and memory by downregulating GSK-3β activity, thereby reversing tau hyperphosphorylation and safeguarding hippocampal neurons from degeneration [102].

To facilitate comparison across preclinical studies and assess translational potential, Table 3 summarizes the principal experimental parameters reported in representative in vivo models, including animal species, dosing, treatment duration, and observed endpoints.

As illustrated, considerable variation exists across studies in dose, route, and duration, making it difficult to directly compare efficacy across disease models or predict human dose equivalence. Nonetheless, a consistent pattern of improved neuronal survival and synaptic integrity emerges across multiple paradigms.

Taken together, these findings highlight the multifaceted neuroprotective potential of exendin-based therapies in preclinical models of AD. By targeting not only memory function and Aβ pathology but also enhancing synaptic plasticity, mitochondrial function, and calcium homeostasis, exendin-4 and exenatide appear to influence several key pathways implicated in neurodegeneration. While these results are encouraging, translating these benefits from animal models to human clinical outcomes remains a critical next step. The complexity of AD pathology calls for further investigation into the mechanisms by which GLP-1RA exert their effects, as well as well-designed clinical trials to assess their safety, efficacy, and potential as adjunct therapies in the treatment of AD.

### 6.2. Safeguarding the Aging Brain: Clinical Evidence Supporting Glp-1ra Therapy

#### 6.2.1. Observational and Pooled Randomized Controlled Trial (RCT) Evidence

Observational data from large patient populations have suggested a strong link between the use of GLP-1RAs and a reduced risk of developing AD. Reports from individuals aged 65 and older, collected through the U. S. Food and Drug Administration (FDA) Adverse Event Reporting System [106], indicated that patients using GLP-1RAs such as exenatide, liraglutide and dulaglutide were less likely to report AD compared to those taking other antidiabetic medications, like metformin. Among these drugs, exenatide appeared to show the strongest association with reduced AD risk, followed by liraglutide and dulaglutide. Further support for the potential cognitive benefits of GLP-1RAs comes from a large-scale study that combined data from multiple randomized, double-blind, placebo-controlled cardiovascular outcome trials involving individuals with type 2 diabetes [107]. This analysis found that those receiving GLP-1RA treatment had a significantly lower incidence of dementia compared to those on placebo. The consistency of these findings across various populations suggests that GLP-1RA therapy could play a protective role against cognitive decline in people with type 2 diabetes.

#### 6.2.2. REWIND Trial—Dulaglutide

Dulaglutide has demonstrated potential in slowing cognitive decline in individuals with type 2 diabetes. In the REWIND trial [108], a large randomized, double-blind, placebo-controlled study, participants receiving weekly dulaglutide showed a reduced risk of significant cognitive deterioration compared to those on placebo. Cognitive function was assessed using the Montreal Cognitive Assessment (MoCA) and the Digit Symbol Substitution Test (DSST), both at baseline and during follow-up. These findings suggest that dulaglutide may offer neuroprotective benefits for patients with type 2 diabetes and elevated cardiovascular risk.

#### 6.2.3. Semaglutide Biomarker/Immune Modulation Study

Several ongoing studies are investigating the potential of semaglutide in Alzheimer’s disease. One study is examining how semaglutide affects the immune system and other biological processes in people with AD, with participants initially randomized to either semaglutide or placebo before all receiving semaglutide in a later phase [109].

#### 6.2.4. EVOKE & EVOKE Plus—Semaglutide Phase III Trials

The EVOKE and EVOKE Plus trials [110] are focused on individuals in the early stages of AD, assessing whether semaglutide can slow cognitive decline. Participants are randomly assigned to receive semaglutide or placebo over a period of several years, with regular clinic visits, cognitive assessments, imaging studies and blood sample collections. The EVOKE and EVOKE+ phase III trials completed recruitment in late 2023 and are now in their long-term follow-up phase, with preliminary safety reports indicating a tolerable adverse-event profile consistent with prior metabolic studies [110]. Although final cognitive endpoints are pending, these studies mark a critical step toward translating GLP-1RA neuroprotective mechanisms into large-scale clinical application.

#### 6.2.5. ISAP Trial—Oral Semaglutide in Preclinical/Prodromal AD

Additionally, the ISAP trial is evaluating oral semaglutide in adults with preclinical or prodromal AD. The study examines its effects on amyloid accumulation, tau pathology, and neuroinflammation. Participants undergo PET and MRI scans, blood tests for biomarkers, and repeated cognitive assessments [111]. The primary goal is to assess changes in tau levels over one year, providing insight into whether semaglutide could help prevent or delay neurodegeneration associated with AD.

#### 6.2.6. Exenatide—Proof-of-Concept in Mild Cognitive Impairment (MCI)

A proof-of-concept study [112] investigated the effects of weekly slow-release exenatide in individuals with MCI, a condition considered an early stage of AD. Over 32 weeks, no significant improvements in cognitive performance were observed compared to the control group. Interestingly, female participants receiving exenatide showed a decline in cognitive scores. While reductions in fasting glucose and body weight were noted, these changes did not impact cognitive outcomes.

#### 6.2.7. Exenatide—18-Month Phase II Trial in Early AD

In a separate 18-month double-blind, placebo-controlled phase II trial in early AD, exenatide was found to be safe and well tolerated, though it did not lead to significant clinical, cognitive, magnetic resonance imaging (MRI), or biomarker improvements [113]. A decrease in plasma neuronal extracellular vesicle Aβ42 levels was observed, but its clinical importance remains unclear. The trial’s early termination limited its ability to demonstrate potential disease-modifying effects.

#### 6.2.8. Liraglutide—Evaluating the Effects of the Novel GLP-1 Analogue Liraglutide in Alzheimer’s Disease (ELAD) Trial

The ELAD trial is among the first multicenter randomized, double-blind, placebo-controlled studies to test a GLP-1 receptor agonist for disease modification in Alzheimer’s disease [114]. Preclinical and translational work supporting liraglutide’s neurobiological rationale points to actions on amyloid, tau, neuroinflammation, synaptic function, and brain glucose handling [115]. ELAD employed a 12-month Phase IIb design in adults with mild Alzheimer’s dementia (without diabetes), randomizing daily subcutaneous liraglutide (up to 1.8 mg) versus placebo with MRI, [18F]FDG-PET, and comprehensive neuropsychological testing at baseline and follow-up [114,116]. Although ELAD did not meet its primary [18F]FDG-PET endpoint of improved cerebral glucose metabolism, this aligns with earlier small RCT data showing no FDG-PET benefit over 26 weeks [117]. Notably, ELAD reported significant secondary/exploratory benefits, including ~50% less regional atrophy (temporal and parietal cortices) and ~18% slower cognitive decline versus placebo over 12 months [116]. Liraglutide was generally well tolerated with predominantly mild gastrointestinal adverse events. Collectively, these findings provide proof-of-concept that GLP-1 analogues may attenuate structural and cognitive deterioration in Alzheimer’s disease and merit larger confirmatory trials.

#### 6.2.9. Liraglutide—Phase IIb (AAIC 2024 and Related Imaging Studies)

Recent findings from a large phase IIb trial presented at the Alzheimer’s Association International Conference 2024 [116] showed that patients with mild to moderate AD who received liraglutide for 12 months had a slower decline in temporal lobe volume, overall grey matter volume, and cortical volume compared to those on placebo. Cognitive assessments also suggested potential benefits in slowing disease progression. A separate study with 38 AD patients found that liraglutide treatment helped maintain cerebral glucose metabolism [117], with the placebo group showing a significant decline. Although changes in cognition and amyloid retention were not statistically significant, the findings point to liraglutide’s potential in preserving brain metabolism. Additional research explored whether liraglutide could improve glucose transport across the blood–brain barrier. In patients with AD, liraglutide significantly increased glucose transfer to levels comparable to healthy individuals, suggesting a restorative effect on impaired glucose transport [118]. The EVOKE and EVOKE+ phase III trials are currently in their main treatment phase, with completion expected in September 2025. A 52-week blinded extension will continue through October 2026 [61,119]. A 2025 pooled safety analysis from prior semaglutide trials in adults aged ≥ 65 years demonstrated a tolerable adverse-event profile, with gastrointestinal events (primarily nausea, vomiting, and diarrhea) being the most common (44.6–73.8%), and discontinuation rates of 9.3–12.4% in older adults compared to 5.7–8.7% in younger populations [104]. Although final cognitive endpoints are still pending and expected by late 2025, these studies mark a critical step toward translating GLP-1RA neuroprotective mechanisms into large-scale clinical application [114]. Earlier studies in cognitively normal individuals with subjective memory complaints showed that liraglutide enhanced neural connectivity within the default mode network (DMN), although no immediate cognitive improvements were seen, indicating the need for longer-term studies [120].

## 7. Limitations and Challenges of GLP-1RAs in Neurodegeneration

### 7.1. Translational Barriers & CNS Target Engagement

The key challenges in exploring GLP-1RAs for neurodegenerative diseases (AD) begin with findings from animal studies. While some transgenic mouse models have shown encouraging results, including reduced Aβ accumulation and improved cognitive performance, other models have failed to demonstrate these effects [121]. This variability highlights the complexity of neurodegenerative diseases (AD and PD) and suggests that therapeutic responses may be highly dependent on disease stage and genetic background [122]. Translating these findings into human clinical studies brings further difficulties. Although GLP-1RAs are known to cross the blood–brain barrier [123], the efficiency of central delivery and the appropriate therapeutic dose needed to achieve neuroprotection in humans remain unclear [124]. Human trials so far did not significantly alter core AD biomarkers (Aβ and tau) nor show improvements in cognition, although metabolic improvements—such as better glucose control and weight reduction—have been observed [66].

A further challenge lies in the biological plausibility and dose–response relationships observed clinically. While preclinical studies show robust neuroprotective effects via anti-inflammatory, anti-apoptotic, and mitochondrial pathways, the extent to which therapeutic doses of GLP-1RAs achieve sufficient central nervous system (CNS) penetration in humans remains uncertain. Liraglutide and semaglutide are large, hydrophilic peptides (3751.2 Da and 4113.6 Da, respectively) with limited blood–brain barrier permeability, and studies using fluorescently labeled compounds demonstrate that semaglutide does not cross the regular blood–brain barrier but rather accesses discrete brain regions through circumventricular organs and sites adjacent to ventricles [125]. Cerebrospinal fluid (CSF) concentrations of liraglutide are minimal even after months of treatment in humans, suggesting very limited passage across the blood–CSF barrier [7,11]. Without direct evidence of receptor occupancy or signaling in human brain tissue, it is difficult to confirm whether observed clinical benefits arise from central GLP-1R engagement or from systemic metabolic and vascular effects.

### 7.2. Tolerability, Adherence, and Safety (And Their Regulatory Implications)

Attrition and adherence issues are another recurrent concern. Gastrointestinal side effects, including nausea and weight loss, are highly prevalent with GLP-1RAs. Systematic analyses indicate that 40–70% of patients experience gastrointestinal adverse events, with some studies reporting rates up to 85% [126,127,128,129,130,131,132,133,134]. Nausea affects up to 50% of patients, vomiting and diarrhea are common, and these symptoms lead to treatment discontinuation in up to 12% of patients compared to approximately 2% with placebo [135], an attrition pattern that may disproportionately exclude frail elderly participants. Consequently, trial populations often become progressively younger and healthier over time, introducing selection bias that inflates apparent cognitive benefit. Parallel limitations exist in observational cohorts, where confounding by indication, socioeconomic status, or concomitant medication use (e.g., metformin, SGLT2 inhibitors) may account for part of the observed risk reduction.

Safety considerations are also paramount. Although GLP-1RAs are generally safe and well tolerated in individuals with type 2 diabetes, older adults with AD may be more prone to side effects due to age and comorbidities. Common adverse effects include gastrointestinal issues such as nausea, vomiting, diarrhea, and delayed gastric emptying, which are often dose-related and tend to decrease over time [136]. Long-acting formulations typically lead to fewer nausea and vomiting complaints but are associated with higher rates of diarrhoea. In rare cases, more serious side effects have been reported, including cutaneous reactions and kidney injury [137]. Additional potential adverse events include hepatic, immunologic, endocrine, metabolic, hematologic, neurological, cardiovascular complications and angioedema [136]. Concerns about possible associations with pancreatitis and certain neoplasms have also been raised, although current clinical evidence is inconclusive. Regulatory authorities such as the FDA and the European Medicines Agency (EMA) currently view these risks as manageable with ongoing monitoring [138]. Injection site reactions and antibody formation occur more frequently with exendin-4-based GLP-1RAs [139]. Given that many of these side effects are already common in elderly individuals, cautious use and vigilant monitoring are essential when considering GLP-1RAs for patients with neurodegenerative diseases.

### 7.3. Methodological Limitations: Power, Endpoints, and Analyses

Despite growing enthusiasm surrounding GLP-1RAs as potential neuroprotective agents, the clinical landscape remains fragmented and methodologically heterogeneous. Most published studies to date are exploratory in nature, characterized by modest sample sizes, short follow-up durations, and considerable variation in patient selection criteria. For instance, the ELAD trial enrolled 204 participants with mild Alzheimer’s disease, and approximately 102 participants per group completed the 52-week treatment phase, limiting statistical power to detect subtle cognitive changes [114]. The trial failed to meet its primary endpoint of changes in cerebral glucose metabolic rate, though secondary outcomes showed some promising signals including reduced brain volume loss. Similarly, early studies investigating exenatide in Parkinson’s disease included 45 patients in an initial open-label pilot trial [140], 62 patients in a Phase 2 trial [141], and 194 participants in the recent Phase 3 Exenatide-PD3 trial, which ultimately found no evidence to support exenatide as a disease-modifying treatment [142]. Such small cohorts raise concerns about both Type I and Type II errors, and the generalizability of results to real-world populations remains uncertain. The divergent endpoints and analytical frameworks across studies further complicate interpretation. Some trials use global cognitive batteries (e.g., ADAS-Cog, CDR-SB), whereas others employ domain-specific or exploratory composite scores, leading to inconsistent effect estimates and difficulties in meta-analysis. Biomarker endpoints also vary, some focusing on amyloid and tau changes, others on FDG-PET metabolic flux or MRI volumetrics [66]. The lack of standardized cognitive and imaging metrics, coupled with differing trial durations (ranging from 6 months to 2 years), undermines cross-trial comparability and reproducibility. Further, many trials employ per-protocol rather than intention-to-treat analyses, thereby excluding participants who discontinue due to adverse events, potentially biasing outcomes toward efficacy.

### 7.4. Population Heterogeneity and Metabolic/Genetic Confounding

Another critical limitation lies in population heterogeneity. Most trials preferentially recruit highly selected patient groups, often younger, with fewer comorbidities, and largely of European ancestry, thereby underrepresenting the ethnically and metabolically diverse populations most affected by neurodegenerative disease. Cognitive reserve, glycemic control, vascular burden, and APOE genotype can each modulate disease progression and drug responsiveness; yet few studies are adequately powered to explore these interactions. Moreover, coexisting diabetes, a common inclusion criterion in earlier GLP-1RA studies, may itself confound neuroprotective effects through independent metabolic and vascular improvements. As a result, it remains uncertain whether observed cognitive benefits are driven by direct central GLP-1R activation or by secondary systemic effects such as improved insulin sensitivity and cerebrovascular perfusion.

### 7.5. Publication Bias and Reporting Transparency in the Evidence Base

Finally, the literature is affected by publication bias and incomplete reporting of null findings. Many smaller investigator-led studies are published following positive interim analyses, while negative or inconclusive data are less widely disseminated. These bias skews meta-analyses toward overestimation of effect size and underestimation of variability. Transparency and preregistration of outcomes, particularly for ongoing large-scale efforts such as EVOKE and EVOKE+ (evaluating semaglutide in over 1800 participants with early Alzheimer’s disease) [110], will be crucial to resolving these uncertainties. Moving forward, the field must prioritize adequately powered, multicentric, biomarker-driven trials with standardized endpoints, diverse patient recruitment, and open data sharing to establish the true therapeutic potential of GLP-1RAs in neurodegenerative disease.

## 8. Future Directions for Glp-1r–Based Therapy in Neurodegenerative Diseases

Advancing GLP-1RAs as treatments for neurodegenerative diseases will require addressing current delivery and efficacy challenges. Improving brain penetration is essential, and nanoparticle-based delivery systems offer a promising solution. Targeted nanocarriers [143,144] can enhance transport across the blood–brain barrier and deliver GLP-1RAs more efficiently to affected regions, potentially reducing systemic side effects. Personalized therapeutic strategies will also be critical. AI-driven tools can help identify patients most likely to benefit from GLP-1RA therapy based on genetic profiles, biomarker data, and neuroimaging results. Moreover, AI can assist in optimizing dosing and exploring combination therapies with agents targeting complementary pathways [145]. Future clinical trials should focus on early-stage or at-risk populations, where interventions may be more effective. Long-term studies are needed to clarify safety and durability of response. Additionally, new delivery approaches—such as intranasal formulations or long-acting implants—could improve adherence and therapeutic consistency [146]. That said, mechanistic and translational gaps are outlined in Table 4 below.

### 8.1. Precision Neuroincretin Medicine

Advances in GLP-1 biology and neurodegeneration now invite a precision “neuroincretin” approach: tailoring GLP-1RA therapy to individual patient biology. GLP-1RAs exert broad anti-inflammatory, metabolic and neurotrophic effects [147], but the heterogeneity of Alzheimer’s, Parkinson’s and Huntington’s disease demands personalized strategies. Emerging multimodal biomarkers, genetic risk (e.g., APOE4, LRRK2, TREM2), metabolic status (insulin resistance, diabetes), imaging (amyloid/tau PET, MRI), and fluid omics (CSF proteomics, metabolomics), could be integrated into companion-diagnostic frameworks as in oncology. For example, patients with evidence of brain insulin resistance or heightened neuroinflammation might be prioritized for GLP-1RA therapy, whereas others might benefit more from alternative approaches. By aligning GLP-1RA modality and dosing to a patient’s molecular profile, a concept we term precision neuroincretin medicine, clinicians could maximize efficacy and minimize off-target effects.

### 8.2. Omics-Driven Patient Stratification

High-throughput omics technologies will drive this precision paradigm by uncovering GLP-1–related signatures. For instance, integrated genomics, transcriptomics, proteomics and metabolomics can identify pathways or biomarkers predictive of GLP-1RA response. Single-cell RNA sequencing in postmortem or induced-pluripotent-stem-cell (iPSC) models of AD, PD and HD can map which neuronal and glial subpopulations express GLP-1R or downstream effectors, and how GLP-1RAs reprogram their states. Recent multi-omic microglia atlases reveal diverse “disease-associated microglia” subtypes (DAM, MGnD, WAM, LDAM) with distinct transcriptional profiles [148,149,150,151]. Future studies could apply single-cell profiling before and after GLP-1RA treatment to see which microglial or astrocyte states are shifted. Likewise, spatial transcriptomics will place these cellular changes in anatomical context, correlating GLP-1RA distribution (by imaging mass cytometry or labeled analogs) with regional gene expression changes. In practice, scRNA-seq and spatial maps from patient brain samples or organoid models could identify GLP-1RA-sensitive pathways (e.g., inflammatory signaling, autophagy genes) as candidate biomarkers. These multimodal data layers should reveal “responder” endophenotypes, turning high-dimensional profiles into actionable patient stratification. Indeed, emerging technologies including single-cell omics and spatial transcriptomics are already being applied to align biomarker profiles with disease states and improve patient stratification. By mining such data, researchers may identify CSF or blood biomarkers (e.g., cytokine panels, metabolite signatures, miRNA changes) that track target engagement and therapeutic effect of GLP-1RAs.

### 8.3. Artificial Intelligence (AI), Predictive Modeling, and Digital Twins

Artificial intelligence will be essential to parse these complex datasets and match patients to therapies. Machine learning classifiers trained on clinical and multi-omic data could predict which individuals are most likely to benefit from GLP-1RA treatment. For example, models could combine polygenic risk scores (APOE, TREM2, HTT repeat length), metabolic measures (HbA1c, insulin levels), and cognitive scores to estimate response probability. Advanced approaches like digital twins, computational models of an individual’s physiology, could simulate how a given GLP-1RA regimen alters neuronal and immune signaling. Indeed, a recent review highlights that AI-guided personalization and digital twins are promising strategies in GLP-1 nanomedicine [152]. In practice, an AI system might analyze a patient’s MRI/PET scans, fluid biomarkers and genomics to recommend a tailored GLP-1RA (or combination of incretin agonists) and dosing schedule. Early clinical trials could use adaptive designs with real-time data monitoring (wearables, cognitive tests) feeding back into AI algorithms to optimize dose. Overall, integrating AI-driven analytics with biomarker profiling could create “closed-loop” precision therapies: algorithms continually refine patient selection and dosing as new data accrue.

### 8.4. Next-Generation CNS Delivery Platforms

Improving CNS delivery of GLP-1RAs remains a key challenge. Nanoparticle platforms are already being engineered for this purpose. Polymeric nanoparticles functionalized with blood–brain barrier (BBB) targeting ligands (such as Angiopep-2) can encapsulate GLP-1RAs and ferry them across the BBB [153,154,155]. Angiopep-2 is a 19-amino acid peptide that binds the low-density lipoprotein receptor-related protein 1 (LRP1), which is widely expressed on brain endothelial cells and activates receptor-mediated transcytosis across the BBB [156]. Such nanocarriers prolong peptide half-life, enable sustained release, and can be tuned to target receptors on neurons or glia. Future nanotherapeutics might co-deliver GLP-1RA with imaging agents (“theranostics”) or sensors, allowing in vivo tracking of biodistribution

### 8.5. Intranasal Route and Other Innovative Delivery Approaches

Intranasal delivery offers another innovative route. By targeting the olfactory/trigeminal pathways, intranasal administration bypasses systemic degradation and minimizes gastrointestinal side effects. Recent rodent studies found that intranasal GLP-1RAs achieve remarkably high brain concentrations: for example, intranasal dulaglutide rapidly accumulates in the hippocampus and neocortex, regions most vulnerable in Alzheimer’s disease, much more efficiently than peripheral delivery. In CD-1 mice and APP/PS1 Alzheimer’s models, intranasal dulaglutide, exenatide, and the dual agonist DA4-JC showed widespread brain distribution with highest uptake in hippocampus and neocortex within 30 min [157]. These findings suggest that nose-to-brain administration of GLP-1RAs could greatly enhance CNS penetration and cognitive outcomes, even if some amyloid pathology is present. Novel intranasal formulations (e.g., cell-penetrating peptide-conjugated GLP-1 NPs) have already shown rapid cognitive benefits in animal models within minutes [158]. In the future, biocompatible “exosome-like” nanovesicles may be loaded with GLP-1 analogs for chronic nasal administration. Other cutting-edge delivery approaches are on the horizon. Implanted osmotic pumps or nanofluidic reservoirs could continuously infuse GLP-1RA into cerebrospinal fluid to maintain stable brain levels. Exosome or virus-based gene therapies delivering GLP-1 or its agonists directly to the brain are also conceivable. Ongoing clinical trials (e.g., EVOKE semaglutide in early AD) will inform how much CNS exposure is needed; positive outcomes may spur adoption of these advanced platforms.

### 8.6. Disease-Specific Precision Applications

#### 8.6.1. Alzheimer’s Disease (AD)

Further, in AD, precision neuroincretin therapy might focus on early stages and mixed pathologies. AD patients with concurrent diabetes or insulin resistance could be prime candidates, as GLP-1RAs improve neuronal insulin signaling. Multi-omic AD cohorts (combining PET imaging of amyloid/tau with transcriptomic profiling) could reveal subgroups with heightened neuroinflammation or synaptic loss, markers that GLP-1RAs might reverse [148,150]. Spatial transcriptomics might map GLP-1R-responsive cells in hippocampus and cortex. Intranasal GLP-1RA delivery, shown to target hippocampal CA1 and prefrontal cortex, could be particularly useful in AD. If ongoing trials like EVOKE show cognitive benefits, that would encourage correlating treatment response with biomarkers (e.g., amyloid load, glial activation, insulin resistance) to refine patient selection.

#### 8.6.2. Parkinson’s Disease (PD)

In PD, the microglial and astrocytic axes are prominent targets. Recent studies demonstrate that GLP-1RAs modulate glial phenotypes: the novel agonist NLY01 prevented microglia from inducing an A1 neurotoxic astrocyte state in α-synuclein PD models [159]. Classic agonists like liraglutide and semaglutide similarly suppressed microglial activation and α-synuclein accumulation in MPTP mouse models, preserving dopaminergic neurons. Clinically, exenatide trials have yielded durable motor and cognitive benefits in PD patients, effects that remarkably persisted for a year after stopping treatment [160]. Future work could use single-cell sequencing of PD patient CSF or even nasal mucosa to identify “incretin-responsive” immune signatures. AI models might integrate gut microbiome data (given gut–brain links in PD) with GLP-1 profiles to further personalize therapy. Longitudinal PET imaging of neuroinflammation (e.g., TSPO tracers) could serve as a biomarker for GLP-1RA efficacy in PD. Overall, leveraging these glial effects in PD through biomarker-guided trials could optimize the use of GLP-1RAs or combined incretin agonists.

#### 8.6.3. Huntington’s Disease (HD)

In HD, metabolic dysfunction is a central feature. HD patients have higher rates of T2D and neurons with mutant huntingtin (mHTT) show impaired insulin signaling. GLP-1RAs may thus target both CNS and peripheral pathology. Preclinical work shows that liraglutide restores insulin sensitivity, boosts cell viability and antioxidant defenses, and activates AMPK-driven autophagy to clear mHTT aggregates [80]. In HD models, these drugs improved motor function and survival while normalizing glucose metabolism. Future precision strategies in HD might stratify patients by metabolic biomarkers (e.g., HOMA-IR, plasma GLP-1 levels) and track response via neuroimaging of striatal volume or functional MRI. Single-cell analysis of HD brains may uncover which neuronal or glial populations benefit most from GLP-1RA. Importantly, an HD “companion diagnostic” might combine huntingtin gene burden with metabolic and oxidative stress markers to guide incretin therapy. Across all these diseases, microglial phenotype programming will be a key theme. GLP-1RAs have been shown to shift microglia away from pro-inflammatory M1/DAM states toward anti-inflammatory or homeostatic phenotypes (sometimes likened to M2/A2) [147]. Future studies should characterize these shifts in patients (for example, measuring cytokines or exosome-derived RNAs). Ultimately, a precision neuroincretin framework combines cutting-edge technologies and insights, multi-omics maps of the diseased brain, AI-driven patient matching, and advanced CNS delivery, to translate GLP-1RAs into effective, individualized neurodegenerative treatments.

## 9. Conclusions

The neuroprotective potential of GLP-1 RAs marks a significant shift in the therapeutic landscape for neurodegenerative diseases, extending well beyond their established role in glucose regulation. GLP-1 signaling has been shown to counteract key pathological processes such as oxidative stress, neuronal apoptosis, mitochondrial dysfunction, and neuroinflammation—all central contributors to conditions like AD, PD, stroke and depression.

The ability of GLP-1R agonists to cross the blood–brain barrier and modulate multiple intracellular pathways, including cAMP/PKA, PI3K/Akt and MAPK, underscores their multifaceted neuroprotective potential. Preclinical models have consistently demonstrated reductions in amyloid-beta deposition, preservation of dopaminergic neurons and attenuation of neuroinflammation, with early clinical trials offering encouraging, though preliminary, support.

However, the translation of these findings into routine clinical use remains challenging. Differences in receptor distribution, disease progression, and treatment responses between animal models and human patients necessitate well-designed, large-scale clinical trials. Establishing optimal dosing, treatment windows, and long-term safety profiles will be essential.

In summary, GLP-1RAs offer a promising and potentially transformative approach to slowing or halting neurodegenerative disease progression. While further rigorous clinical investigation is required, their ability to target fundamental mechanisms of neuronal decline positions them at the forefront of future neuroprotective therapies.

## Figures and Tables

**Figure 1 ijms-26-10743-f001:**
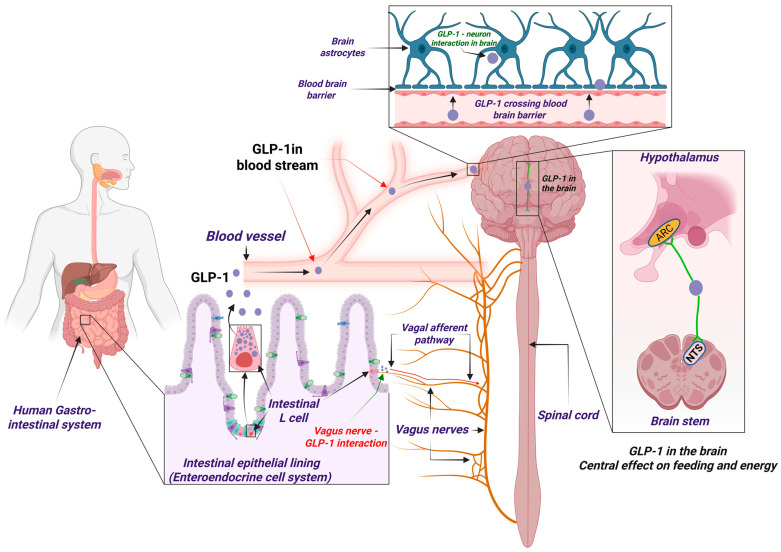
Regulation of Feeding and Energy via Gut–Brain GLP-1 Pathways. The figure above illustrates GLP-1 secretion from enteroendocrine L cells and its actions through humoral and neural pathways. GLP-1 released in the gut may either activate vagal nerves in the intestinal epithelial cells or travel through the humoral route, crossing the blood–brain barrier to interact with neurons in the brain. In the brain, GLP-1 engages with arcuate nucleus neurons (ARC) in the hypothalamus or neurons of the nucleus tractus solitarius (NTS) in the brainstem to regulate feeding and energy.

**Figure 2 ijms-26-10743-f002:**
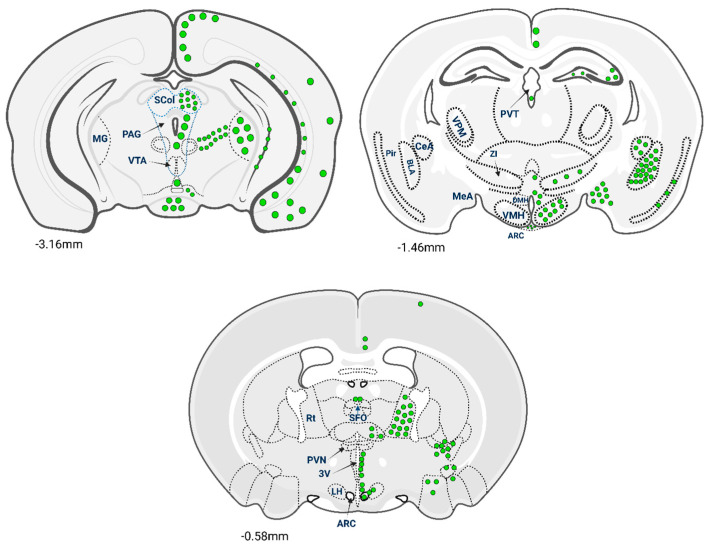
Diagrams of coronal sections showing the distribution of GLP-1R-expressing cell bodies in the mouse brain. Green circles represent the presence of GLP-1R immunoreactive somata (neuronal cell body). The density of the green circles indicates the relative density of the GLP-R-positive somata in each brain region. Brain-section diagrams are based on the Paxinos Mouse Brain Atlas and numerical values next to each section indicate the rostro-caudal position in relation to Bregma. SCol, superior colliculus; PAG, periaqueductal grey area; MG, medial geniculate nucleus; PAG, periaqueductal grey area; VTA, ventral tegmental area; PVT, thalamic paraventricular nucleus; VPM, ventral posteromedial thalamic nucleus; Pir, piriform cortex; CeA, central amygdala; BLA, basolateral amygdala; MeA, medial amygdala; DMH, dorsomedial hypothalamus; VMH, ventromedial hypothalamus; ARC, arcuate nucleus; SFO, subfornical organ; Rt, reticular nucleus; LH, lateral hypothalamus; 3V, Third Ventricle. The illustration’s concept was primarily derived from the data presented in the article [16].

**Figure 3 ijms-26-10743-f003:**
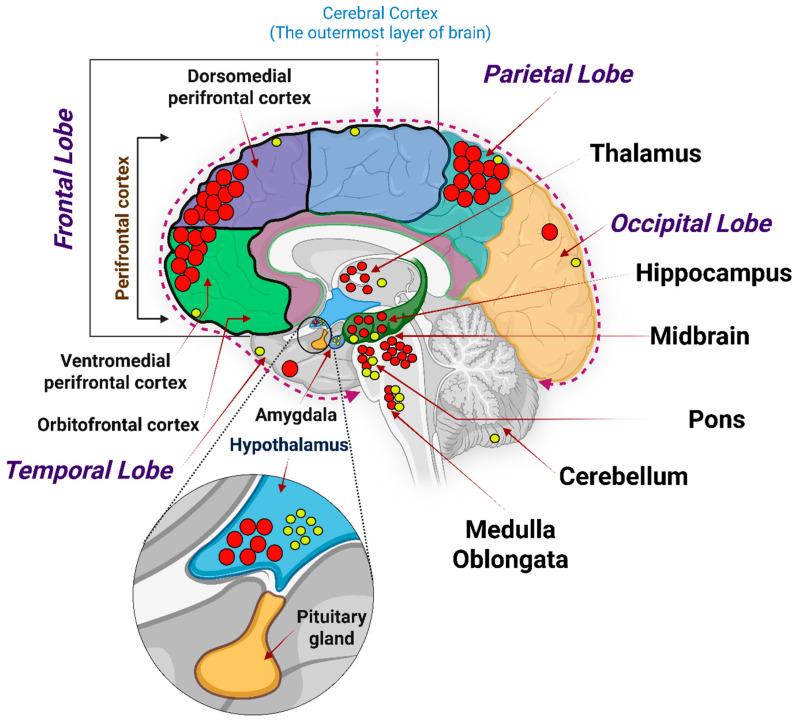
The GLP-1R mRNA expression in different regions of human brain. The red and yellow circles indicate the localization of GLP-1R mRNA based on the data presented in the article [18] and The Human Protein Atlas (HPA), respectively. While the density of the red circles indicates the level of expression of the GLP-1R mRNA in the specific brain region; the number of the yellow circles indicates the approximate transcripts per million (nTPM) mentioned in the HPA. Illustration was created using Biorender.com.

**Figure 4 ijms-26-10743-f004:**
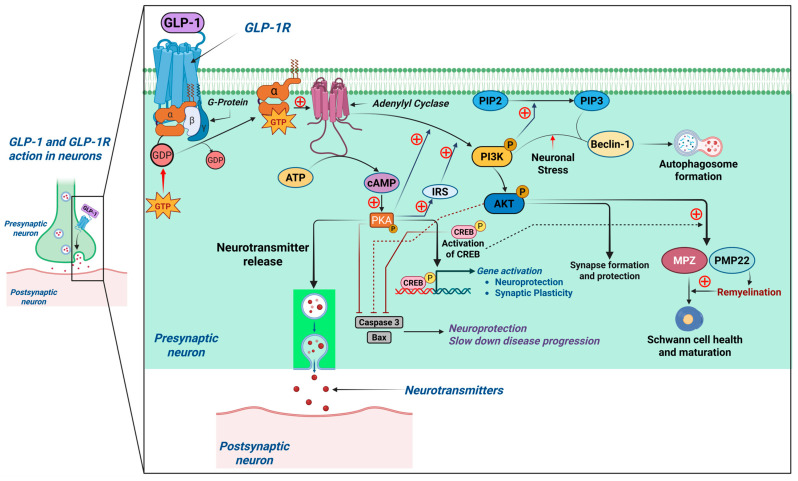
Mechanism of GLP-1 Mediated Neuroprotection. The interactions between GLP-1 and its receptor GLP-1R in presynaptic neurons are depicted in the smaller image on the left. The black rectangle inset is magnified on the right to illustrate the intracellular signaling cascades that result in neuroprotection. GDP—Guanosine diphosphate; GTP—Guanosine triphosphate; cAMP—Cyclic adenosine monophosphate; PIP2—Phosphatidylinositol 4,5-bisphosphate; PIP3—Phosphatidylinositol (3,4,5)-trisphosphate; IRS—Insulin Receptor Substrate; MPZ—Myelin protein zero; PMP22—Peripheral myelin protein 22.

**Table 1 ijms-26-10743-t001:** Expression of GLP-1Rs in different parts of the brain seen in humans, non-human primates and rodents.

Brain Area	Human [Relevant Reference [18]]	Non-Human Primate [Relevant Reference [3]]	Rodents [Relevant Reference [36]]
Cerebral Cortex	Present in all areas except in orbitofrontal cortex	Absent	Absent, except in mouse prefrontal cortex
Hippocampus	Present	Present	Present
Cerebellum	Absent	Present	Present
Hypothalamus	Present	Most abundant expression	Most abundant expression
Amygdala, Thalamus and Brain Stem	Present	Present	Present

**Table 3 ijms-26-10743-t003:** Comparison of principal experimental parameters reported in representative in vivo models of AD, PD, ALS, and HD.

Disease Model	Species/Strain	Compound/Formulation	Dose & Route	Treatment Duration	Main Outcomes	Reference
Alzheimer’s disease (APP/PS1)	Mouse	Liraglutide (Victoza^®^, i.p.)	25 nmol/kg daily, i.p.	8–12 weeks	↓ Aβ plaque load; ↑ synaptic density; improved Morris water maze performance	[59]
Alzheimer’s disease (3xTg-AD)	Mouse	Exenatide (Byetta^®^, i.p.)	0.1 μg/g (100 μg/kg) daily	9 months	No effects on memory performance, Aβ or tau pathology in 3xTg-AD mice	[61]
Parkinson’s disease (MPTP-induced)	Mouse	Exenatide (synthetic, i.p.)	10 µg/kg daily, i.p.	7 days	Preserved TH+ neurons; improved rotarod and open-field scores	[103]
Parkinson’s disease (6-OHDA-induced)	Rat	PT320 (sustained-release exenatide, s.c.)	100 mg/kg weekly (containing 2 mg/kg exendin-4)	3 weeks	↓ L-DOPA-induced dyskinesia; normalized DA turnover	[104]
Amyotrophic lateral sclerosis (SOD1^G93A)	Mouse	Liraglutide, i.p.	25 nmol/kg daily i.p.	From 50 days of age until endpoint	No effect on disease progression, motor neuron counts, glial activation, or survival	[105]
Huntington’s disease (R6/2)	Mouse	Exendin-4, i.p.	10 µg/kg daily	8–12 weeks	Improved motor coordination; ↓ neuronal inclusion bodies; extended survival	[82]

↑ indicates increase and ↓ indicates decrease of levels.

**Table 4 ijms-26-10743-t004:** Outstanding Mechanistic Questions and Translational Barriers that Must Be Addressed to Realize the Full Therapeutic Potential of GLP-1 Receptor Agonists in Neurodegenerative Disease.

Knowledge Gap	Scientific/Clinical Challenge	Proposed Future Direction
Blood–brain barrier penetration and receptor occupancy	Limited in vivo evidence confirming central target engagement in humans	Apply radiolabeled ligand PET and CSF biomarker studies in early-phase trials
Species-specific receptor distribution	Rodent–human divergence in GLP-1R localization and signaling bias	Employ humanized or non-human primate models; correlate expression with pharmacodynamics
Heterogeneity in clinical outcomes	Variation in disease stage, comorbid diabetes, and APOE genotype	Stratified trial designs and AI-driven patient selection
Biomarker inconsistency	Weak linkage between imaging, CSF biomarkers, and cognition	Harmonize biomarker endpoints and integrate multi-modal readouts
Long-term safety in non-diabetic elderly populations	Possible GI, cardiovascular, or weight-loss adverse effects	Longitudinal registries and real-world pharmacovigilance
Translational modeling	Preclinical success not reliably predictive of human efficacy	Systems-level modeling of neuro-metabolic networks; combination therapy trials
Diversity and generalizability	Underrepresentation of non-European populations in trials	Expand recruitment across global cohorts to ensure external validity

## Data Availability

No new data were created or analyzed in this study. Data sharing is not applicable to this article.

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
