# Peer review of "GLP-1 and the Degenerating Brain: Exploring Mechanistic Insights and Therapeutic Potential"

_ijms, 2025, doi:10.3390/ijms262110743_

Round 1

Reviewer 1 Report

Comments and Suggestions for Authors

GLP-1 (glucagon-like peptide-1) plays an important role in the context of neurodegeneration and potential therapies. This review consolidates the mechanistic basis and translational potential of GLP-1RA receptors in age-related neurodegenerative diseases, highlighting both their promising value and the challenges that need to be addressed in future clinical applications.

This work provides a broad approach to the topic, supported by current research findings. The relevance of this topic is supported by numerous publications, including review articles, addressing similar issues:

Ibrahim R, Kambal A, Abdelmajeed MA. Glucagon-Like Peptide-1 Receptor Agonists in Neurodegenerative Diseases: A Comprehensive Review. Cureus. 2025 Sep 16;17(9):e92441. doi: 10.7759/cureus.92441

Hong, CT., Chen, JH. & Hu, CJ. Role of glucagon-like peptide-1 receptor agonists in Alzheimer’s disease and Parkinson’s disease. J Biomed Sci 31, 102 (2024). https://doi.org/10.1186/s12929-024-01090-x

Kopp KO, Glotfelty EJ, Li Y, Greig NH. Glucagon-like peptide-1 (GLP-1) receptor agonists and neuroinflammation: Implications for neurodegenerative disease treatment. Pharmacol Res. 2022;186:106550. doi:10.1016/j.phrs.2022.106550

In my opinion, the work is conscientiously written and takes into account the topic's relevance. However, there is no clear indication of how the proposed manuscript differs from existing review articles on the topic, which would allow us to assess its innovative nature.

Author Response

Reviewer 1

Comment 1:

GLP-1 (glucagon-like peptide-1) plays an important role in the context of neurodegeneration and potential therapies. This review consolidates the mechanistic basis and translational potential of GLP-1RA receptors in age-related neurodegenerative diseases, highlighting both their promising value and the challenges that need to be addressed in future clinical applications.

This work provides a broad approach to the topic, supported by current research findings. The relevance of this topic is supported by numerous publications, including review articles, addressing similar issues:

Ibrahim R, Kambal A, Abdelmajeed MA. Glucagon-Like Peptide-1 Receptor Agonists in Neurodegenerative Diseases: A Comprehensive Review. Cureus. 2025 Sep 16;17(9):e92441. doi: 10.7759/cureus.92441

Hong, CT., Chen, JH. & Hu, CJ. Role of glucagon-like peptide-1 receptor agonists in Alzheimer’s disease and Parkinson’s disease. J Biomed Sci 31, 102 (2024). https://doi.org/10.1186/s12929-024-01090-x

Kopp KO, Glotfelty EJ, Li Y, Greig NH. Glucagon-like peptide-1 (GLP-1) receptor agonists and neuroinflammation: Implications for neurodegenerative disease treatment. Pharmacol Res. 2022;186:106550. doi:10.1016/j.phrs.2022.106550

In my opinion, the work is conscientiously written and takes into account the topic's relevance. However, there is no clear indication of how the proposed manuscript differs from existing review articles on the topic, which would allow us to assess its innovative nature.

Response to the comment 1 of reviewer 1:

We sincerely thank the reviewer for this insightful comment highlighting the importance of clarifying our manuscript’s novelty relative to prior reviews.

Firstly, our review uniquely provides a visual, region-by-region map of GLP-1R distribution in both mouse and human brain, collating very recent human brain mapping data together with the Human Protein Atlas (Brain Atlas) transcriptomic dataset (nTPM) and harmonizing them into a single schematic (Figure 3) and an annotated mouse diagram (Figure 2). While one of the reviewer-cited articles mentions GLP-1R expression data, to our knowledge none of the three reviews (Ibrahim et al., 2025; Hong & Chen, 2024; Kopp et al., 2022) (i) integrate both sources side-by-side, and (ii) deep-outline cortical, limbic, hypothalamic, and brainstem subregions at the diagram level with quantitative cues (HPA nTPM) and species contrasts. This visual synthesis is a major differentiator of our manuscript and directly serves translation by spotlighting species-specific targets and gaps for PET occupancy studies and trial site selection.

Secondly, we have substantially expanded the manuscript to introduce a distinct conceptual framework and forward-looking perspective that differentiates our work from existing literature. Specifically, we added integrated subsections [Under the Section 8; subsections a to f (page 24-28; Line 906 – 1042 in the revised manuscript)] that proposes the precision neuroincretin medicine paradigm, a novel approach aligning GLP-1RA therapy with individualized molecular, metabolic, and genetic profiles across neurodegenerative diseases.

This new section introduces several innovative aspects not covered in prior reviews:

  1. Precision stratification framework: We propose integrating multimodal biomarkers (genetic risk factors such as APOE4, LRRK2, TREM2; metabolic status; fluid omics; and neuroimaging) into patient selection models for GLP-1RA therapy, paralleling approaches in precision oncology.
  2. Application of emerging technologies: We highlight how single-cell RNA sequencing, spatial transcriptomics, and multi-omic integration can uncover GLP-1RA-responsive cell types and pathways, guiding personalized treatment strategies.
  3. AI and digital twin integration: We emphasize that artificial intelligence and computational patient “digital twins” can optimize GLP-1RA dosing, predict response, and enable adaptive clinical trial design, an emerging direction recently noted as promising in GLP-1 nanomedicine.
  4. Next-generation delivery systems: We discuss innovative CNS delivery platforms including BBB-targeted nanoparticles and intranasal formulations, supported by recent preclinical data demonstrating enhanced brain uptake and therapeutic efficacy.
  5. Disease-specific applications: We extend the discussion to precision approaches tailored to Alzheimer’s, Parkinson’s, and Huntington’s diseases, linking disease-specific pathophysiology with GLP-1RA mechanisms and biomarker-guided treatment optimization.

Together, these additions explicitly position the manuscript as a forward-looking synthesis that moves beyond traditional mechanistic reviews by outlining how GLP-1 biology, omics technologies, AI, and advanced delivery systems converge toward a precision medicine paradigm in neurodegeneration. Please see the additions below (Line 906 – 1042 in the revised manuscript):

a. Precision Neuroincretin Medicine

Advances in GLP‐1 biology and neurodegeneration now invite a precision “neuroincretin” approach: tailoring GLP‐1RA therapy to individual patient biology. GLP‐1RAs exert broad anti‐inflammatory, metabolic and neurotrophic effects (149), but the heterogeneity of Alzheimer’s, Parkinson’s and Huntington’s disease demands personalized strategies. Emerging multimodal biomarkers, genetic risk (e.g. APOE4, LRRK2, TREM2), metabolic status (insulin resistance, diabetes), imaging (amyloid/tau PET, MRI), and fluid omics (CSF proteomics, metabolomics), could be integrated into companion‐diagnostic frameworks as in oncology. For example, patients with evidence of brain insulin resistance or heightened neuroinflammation might be prioritized for GLP‐1RA therapy, whereas others might benefit more from alternative approaches. By aligning GLP‐1RA modality and dosing to a patient’s molecular profile, a concept we term precision neuroincretin medicine, clinicians could maximize efficacy and minimize off‐target effects.

b. Omics-Driven Patient Stratification

High‐throughput omics technologies will drive this precision paradigm by uncovering GLP‐1–related signatures. For instance, integrated genomics, transcriptomics, proteomics and metabolomics can identify pathways or biomarkers predictive of GLP‐1RA response. Single‐cell RNA sequencing in postmortem or induced‐pluripotent‐stem‐cell (iPSC) models of AD, PD and HD can map which neuronal and glial subpopulations express GLP‐1R or downstream effectors, and how GLP‐1RAs reprogram their states. Recent multi‐omic microglia atlases reveal diverse “disease‐associated microglia” subtypes (DAM, MGnD, WAM, LDAM) with distinct transcriptional profiles (150-153). Future studies could apply single‐cell profiling before and after GLP‐1RA treatment to see which microglial or astrocyte states are shifted. Likewise, spatial transcriptomics will place these cellular changes in anatomical context, correlating GLP‐1RA distribution (by imaging mass cytometry or labeled analogs) with regional gene expression changes. In practice, scRNA‐seq and spatial maps from patient brain samples or organoid models could identify GLP‐1RA‐sensitive pathways (e.g. inflammatory signaling, autophagy genes) as candidate biomarkers. These multimodal data layers should reveal “responder” endophenotypes, turning high‐dimensional profiles into actionable patient stratification. Indeed, emerging technologies including single‐cell omics and spatial transcriptomics are already being applied to align biomarker profiles with disease states and improve patient stratification. By mining such data, researchers may identify CSF or blood biomarkers (e.g. cytokine panels, metabolite signatures, miRNA changes) that track target engagement and therapeutic effect of GLP‐1RAs.

c. Artificial intelligence (AI), Predicting Modeling, and Digital Twins

Artificial intelligence will be essential to parse these complex datasets and match patients to therapies. Machine learning classifiers trained on clinical and multi‐omic data could predict which individuals are most likely to benefit from GLP‐1RA treatment. For example, models could combine polygenic risk scores (APOE, TREM2, HTT repeat length), metabolic measures (HbA1c, insulin levels), and cognitive scores to estimate response probability. Advanced approaches like digital twins, computational models of an individual’s physiology, could simulate how a given GLP‐1RA regimen alters neuronal and immune signaling. Indeed, a recent review highlights that AI-guided personalization and digital twins are promising strategies in GLP‐1 nanomedicine (154). In practice, an AI system might analyze a patient’s MRI/PET scans, fluid biomarkers and genomics to recommend a tailored GLP‐1RA (or combination of incretin agonists) and dosing schedule. Early clinical trials could use adaptive designs with real‐time data monitoring (wearables, cognitive tests) feeding back into AI algorithms to optimize dose. Overall, integrating AI-driven analytics with biomarker profiling could create “closed-loop” precision therapies: algorithms continually refine patient selection and dosing as new data accrue.

d. Next-Generation CNS Delivery Platforms

Improving CNS delivery of GLP‐1RAs remains a key challenge. Nanoparticle platforms are already being engineered for this purpose. Polymeric nanoparticles functionalized with blood–brain barrier (BBB) targeting ligands (such as Angiopep‐2) can encapsulate GLP‐1RAs and ferry them across the BBB (155-157). Angiopep-2 is a 19-amino acid peptide that binds the low-density lipoprotein receptor-related protein 1 (LRP1), which is widely expressed on brain endothelial cells and activates receptor-mediated transcytosis across the BBB (158). Such nanocarriers prolong peptide half‐life, enable sustained release, and can be tuned to target receptors on neurons or glia. Future nanotherapeutics might co‐deliver GLP‐1RA with imaging agents ("theranostics") or sensors, allowing in vivo tracking of biodistribution

e. Intranasal Route and Other Innovative Delivery Approaches

Intranasal delivery offers another innovative route. By targeting the olfactory/trigeminal pathways, intranasal administration bypasses systemic degradation and minimizes gastrointestinal side effects. Recent rodent studies found that intranasal GLP‐1RAs achieve remarkably high brain concentrations: for example, intranasal dulaglutide rapidly accumulates in the hippocampus and neocortex, regions most vulnerable in Alzheimer’s disease, much more efficiently than peripheral delivery. In CD-1 mice and APP/PS1 Alzheimer's models, intranasal dulaglutide, exenatide, and the dual agonist DA4-JC showed widespread brain distribution with highest uptake in hippocampus and neocortex within 30 minutes (159). These findings suggest that nose‐to‐brain administration of GLP‐1RAs could greatly enhance CNS penetration and cognitive outcomes, even if some amyloid pathology is present. Novel intranasal formulations (e.g. cell‐penetrating peptide‐conjugated GLP‐1 NPs) have already shown rapid cognitive benefits in animal models within minutes (160). In the future, biocompatible “exosome‐like” nanovesicles may be loaded with GLP‐1 analogs for chronic nasal administration. Other cutting‐edge delivery approaches are on the horizon. Implanted osmotic pumps or nanofluidic reservoirs could continuously infuse GLP‐1RA into cerebrospinal fluid to maintain stable brain levels. Exosome or virus‐based gene therapies delivering GLP‐1 or its agonists directly to the brain are also conceivable. Ongoing clinical trials (e.g. EVOKE semaglutide in early AD) will inform how much CNS exposure is needed; positive outcomes may spur adoption of these advanced platforms.

f. Disease-Specific Precision Applications

i. Alzheimer’s Diseases (AD)

Further, in AD, precision neuroincretin therapy might focus on early stages and mixed pathologies. AD patients with concurrent diabetes or insulin resistance could be prime candidates, as GLP‐1RAs improve neuronal insulin signaling. Multi‐omic AD cohorts (combining PET imaging of amyloid/tau with transcriptomic profiling) could reveal subgroups with heightened neuroinflammation or synaptic loss, markers that GLP‐1RAs might reverse (150, 152). Spatial transcriptomics might map GLP‐1R-responsive cells in hippocampus and cortex. Intranasal GLP‐1RA delivery, shown to target hippocampal CA1 and prefrontal cortex, could be particularly useful in AD. If ongoing trials like EVOKE show cognitive benefits, that would encourage correlating treatment response with biomarkers (e.g. amyloid load, glial activation, insulin resistance) to refine patient selection.

ii. Parkinson’s Diseases (PD)

In PD, the microglial and astrocytic axes are prominent targets. Recent studies demonstrate that GLP‐1RAs modulate glial phenotypes: the novel agonist NLY01 prevented microglia from inducing an A1 neurotoxic astrocyte state in α‐synuclein PD models (161). Classic agonists like liraglutide and semaglutide similarly suppressed microglial activation and α‐synuclein accumulation in MPTP mouse models, preserving dopaminergic neurons. Clinically, exenatide trials have yielded durable motor and cognitive benefits in PD patients, effects that remarkably persisted for a year after stopping treatment (162). Future work could use single‐cell sequencing of PD patient CSF or even nasal mucosa to identify “incretin‐responsive” immune signatures. AI models might integrate gut microbiome data (given gut–brain links in PD) with GLP‐1 profiles to further personalize therapy. Longitudinal PET imaging of neuroinflammation (e.g. TSPO tracers) could serve as a biomarker for GLP‐1RA efficacy in PD. Overall, leveraging these glial effects in PD through biomarker‐guided trials could optimize use of GLP‐1RAs or combined incretin agonists.

iii.       Huntington’s Disease (HD)

In HD, metabolic dysfunction is a central feature. HD patients have higher rates of T2D and neurons with mutant huntingtin (mHTT) show impaired insulin signaling. GLP‐1RAs may thus target both CNS and peripheral pathology. Preclinical work shows that liraglutide restores insulin sensitivity, boosts cell viability and antioxidant defenses, and activates AMPK‐driven autophagy to clear mHTT aggregates (81). In HD models, these drugs improved motor function and survival while normalizing glucose metabolism. Future precision strategies in HD might stratify patients by metabolic biomarkers (e.g. HOMA‐IR, plasma GLP‐1 levels) and track response via neuroimaging of striatal volume or functional MRI. Single‐cell analysis of HD brains may uncover which neuronal or glial populations benefit most from GLP‐1RA. Importantly, an HD “companion diagnostic” might combine huntingtin gene burden with metabolic and oxidative stress markers to guide incretin therapy. Across all these diseases, microglial phenotype programming will be a key theme. GLP‐1RAs have been shown to shift microglia away from pro‐inflammatory M1/DAM states toward anti‐inflammatory or homeostatic phenotypes (sometimes likened to M2/A2) (149). Future studies should characterize these shifts in patients (for example, measuring cytokines or exosome‐derived RNAs). Ultimately, a precision neuroincretin framework combines cutting‐edge technologies and insights, multi‐omics maps of the diseased brain, AI‐driven patient matching, and advanced CNS delivery, to translate GLP‐1RAs into effective, individualized neurodegenerative treatments.

Reviewer 2 Report

Comments and Suggestions for Authors

This manuscript provides a comprehensive and timely overview of the neuroprotective mechanisms of glucagon-like peptide-1 (GLP-1) and its receptor (GLP-1R) in neurodegenerative diseases, particularly Alzheimer’s and Parkinson’s disease. The topic is of significant scientific and clinical relevance, and the review is generally well-organized and informative. The manuscript would, however, benefit from several targeted revisions to enhance its clarity, depth, and overall scholarly impact:

Update the literature and clinical data:

Please include the most recent findings from the EVOKE and ELAD clinical trials, which have reported new results since 2024. Incorporating these updates will ensure the review reflects the most current progress in the field (lines 27–31, 35).

Strengthen the critical discussion of clinical evidence:

The current version tends to highlight positive results. It would be valuable to include a more balanced evaluation of clinical studies, discussing limitations such as sample size, patient diversity, and heterogeneity in trial design (lines 268–291, 357–360).

Expand discussion on species differences and translational relevance:

The section on GLP-1R expression across species (lines 122–151) could be further developed to emphasize the implications of interspecies variability for translational research. Explicitly acknowledge the limitations of extrapolating animal data to human disease mechanisms.

Add a concise summary of research gaps and future directions:

A summary table highlighting unresolved mechanistic questions and potential avenues for further research (lines 226–232, 387–402) would improve readability and underscore the review’s contribution to advancing the field.

Refine the writing for clarity and consistency:

A final round of English editing is recommended to enhance conciseness, standardize abbreviations, and improve overall readability (lines 1–11).

Overall, this is a solid and well-researched review with high relevance to the therapeutics of neurodegenerative diseases. However, it will be accepted before a minor revision to the standard of IJMS.

Author Response

Reviewer 2

Comments and Suggestions for Authors

This manuscript provides a comprehensive and timely overview of the neuroprotective mechanisms of glucagon-like peptide-1 (GLP-1) and its receptor (GLP-1R) in neurodegenerative diseases, particularly Alzheimer’s and Parkinson’s disease. The topic is of significant scientific and clinical relevance, and the review is generally well-organized and informative. The manuscript would, however, benefit from several targeted revisions to enhance its clarity, depth, and overall scholarly impact:

Comment 1:

Update the literature and clinical data: Please include the most recent findings from the EVOKE and ELAD clinical trials, which have reported new results since 2024. Incorporating these updates will ensure the review reflects the most current progress in the field (lines 27–31, 35).

Response to the comment 1:

Thank you for this comment. In response to the comment of the reviewer we have now described all the clinical evidence supporting GLP-1RAs in neurodegenerative disease under section 6, subsections b. The most recent findings of EVOKE and EVOKE+ trials have been added in sub-section b (iv) (in page 20, Line 695-703) and in sub-section b (ix) (line 762-770). The new addition on EVOKE now reads,

  1. EVOKE & EVOKE Plus — Semaglutide Phase III Trials

The EVOKE and EVOKE Plus trials (111) are focused on individuals in the early stages of AD, assessing whether semaglutide can slow cognitive decline. Participants are randomly assigned to receive semaglutide or placebo over a period of several years, with regular clinic visits, cognitive assessments, imaging studies and blood sample collections. The EVOKE and EVOKE+ phase III trials completed recruitment in late 2023 and are now in their long-term follow-up phase, with preliminary safety reports indicating a tolerable adverse-event profile consistent with prior metabolic studies (111). Although final cognitive endpoints are pending, these studies mark a critical step toward translating GLP-1RA neuroprotective mechanisms into large-scale clinical application.

In subsection b (iv) (in page 20, Line 695-703)

The EVOKE and EVOKE+ phase III trials are currently in their main treatment phase, with completion expected in September 2025. A 52-week blinded extension will continue through October 2026 (23640454, 35140584). A 2025 pooled safety analysis from prior semaglutide trials in adults aged ≥65 years demonstrated a tolerable adverse-event profile, with gastrointestinal events (primarily nausea, vomiting, and diarrhea) being the most common (44.6%-73.8%), and discontinuation rates of 9.3%-12.4% in older adults compared to 5.7%-8.7% in younger populations (32848559). Although final cognitive endpoints are still pending and expected by late 2025, these studies mark a critical step toward translating GLP-1RA neuroprotective mechanisms into large-scale clinical application (100).

The following sub-section b (viii) has been added for the ELAD trial (page 21, line 730-747) which now reads:

viii.     Liraglutide - Evaluating the effects of the novel GLP-1 analogue lirag-lutide in Alzheimer's disease (ELAD) Trial

The ELAD trial is among the first multicenter randomized, double-blind, placebo-controlled studies to test a GLP-1 receptor agonist for disease modification in Alzheimer’s disease (115). Preclinical and translational work supporting liraglutide’s neurobiological rationale points to actions on amyloid, tau, neuroinflammation, synaptic function, and brain glucose handling (116). ELAD employed a 12-month Phase IIb design in adults with mild Alzheimer’s dementia (without diabetes), randomizing daily subcutaneous liraglutide (up to 1.8 mg) versus placebo with MRI, [18F]FDG-PET, and comprehensive neuropsychological testing at baseline and follow-up (115, 117). Although ELAD did not meet its primary [18F]FDG-PET endpoint of improved cerebral glucose metabolism, this aligns with earlier small RCT data showing no FDG-PET benefit over 26 weeks (118). Notably, ELAD reported significant secondary/exploratory benefits, including ~50% less regional atrophy (temporal and parietal cortices) and ~18% slower cognitive decline versus placebo over 12 months (117). Liraglutide was generally well tolerated with predominantly mild gastrointestinal adverse events. Collectively, these findings provide proof-of-concept that GLP-1 analogues may attenuate structural and cognitive deterioration in Alzheimer’s disease and merit larger confirmatory trials.

Comment 2:

Strengthen the critical discussion of clinical evidence: The current version tends to highlight positive results. It would be valuable to include a more balanced evaluation of clinical studies, discussing limitations such as sample size, patient diversity, and heterogeneity in trial design (lines 268–291, 357–360).

Response to the comment 1:

Thank you for this comment. To address this comment, we have re-written section 7 (now “Limitations and Challenges of GLP-1RAs in Neurodegeneration”) with multiple sub-sections (a - e). The section 7 now reads as follows (page 22 to 24, Line 775-886 in the revised manuscript).

7. Limitations and Challenges of GLP-1RAs in Neurodegeneration

a. Translational Barriers & CNS Target Engagement

The key challenges in exploring GLP-1RAs for neurodegenerative diseases (AD) begin with findings from animal studies. While some transgenic mouse models have shown encouraging results, including reduced Aβ accumulation and improved cognitive performance, other models have failed to demonstrate these effects (123). This variability highlights the complexity of neurodegenerative diseases (AD and PD) and suggests that therapeutic responses may be highly dependent on disease stage and genetic background (124). Translating these findings into human clinical studies brings further difficulties. Although GLP-1RAs are known to cross the blood–brain barrier  (125), the efficiency of central delivery and the appropriate therapeutic dose needed to achieve neuroprotection in humans remain unclear (126). Human trials so far did not significantly alter core AD biomarkers (Aβ and tau) nor show improvements in cognition, although metabolic improvements — such as better glucose control and weight reduction — have been observed (66).

A further challenge lies in the biological plausibility and dose-response relationships observed clinically. While preclinical studies show robust neuroprotective effects via anti-inflammatory, anti-apoptotic, and mitochondrial pathways, the extent to which therapeutic doses of GLP-1RAs achieve sufficient central nervous system (CNS) penetration in humans remains uncertain. Liraglutide and semaglutide are large, hydrophilic peptides (3751.2 Da and 4113.6 Da, respectively) with limited blood–brain barrier permeability, and studies using fluorescently labeled compounds demonstrate that semaglutide does not cross the regular blood-brain barrier but rather accesses discrete brain regions through circumventricular organs and sites adjacent to ventricles (127). Cerebrospinal fluid (CSF) concentrations of liraglutide are minimal even after months of treatment in humans, suggesting very limited passage across the blood-CSF barrier (7, 11). Without direct evidence of receptor occupancy or signaling in human brain tissue, it is difficult to confirm whether observed clinical benefits arise from central GLP-1R engagement or from systemic metabolic and vascular effects.

b. Tolerability, Adherence, and Safety (and Their Regulatory Implications)

Attrition and adherence issues are another recurrent concern. Gastrointestinal side effects, including nausea and weight loss, are highly prevalent with GLP-1RAs. Systematic analyses indicate that 40-70% of patients experience gastrointestinal adverse events, with some studies reporting rates up to 85% (128-136). Nausea affects up to 50% of patients, vomiting and diarrhea are common, and these symptoms lead to treatment discontinuation in up to 12% of patients compared to approximately 2% with placebo (137), an attrition pattern that may disproportionately exclude frail elderly participants. Consequently, trial populations often become progressively younger and healthier over time, introducing selection bias that inflates apparent cognitive benefit. Parallel limitations exist in observational cohorts, where confounding by indication, socioeconomic status, or concomitant medication use (e.g., metformin, SGLT2 inhibitors) may account for part of the observed risk reduction.

Safety considerations are also paramount. Although GLP-1RAs are generally safe and well tolerated in individuals with type 2 diabetes, older adults with AD may be more prone to side effects due to age and comorbidities. Common adverse effects include gastrointestinal issues such as nausea, vomiting, diarrhea, and delayed gastric emptying, which are often dose-related and tend to decrease over time (138). Long-acting formulations typically lead to fewer nausea and vomiting complaints but are associated with higher rates of diarrhoea. In rare cases, more serious side effects have been reported, including cutaneous reactions and kidney injury (139). Additional potential adverse events include hepatic, immunologic, endocrine, metabolic, hematologic, neurological, cardiovascular complications and angioedema (138). Concerns about possible associations with pancreatitis and certain neoplasms have also been raised, although current clinical evidence is inconclusive. Regulatory authorities such as the FDA and the European Medicines Agency (EMA) currently view these risks as manageable with ongoing monitoring (140). Injection site reactions and antibody formation occur more frequently with exendin-4-based GLP-1RAs (141). Given that many of these side effects are already common in elderly individuals, cautious use and vigilant monitoring are essential when considering GLP-1RAs for patients with neurodegenerative diseases.

c. Methodological Limitations: Power, Endpoints, and Analyses

Despite growing enthusiasm surrounding GLP-1RAs as potential neuroprotective agents, the clinical landscape remains fragmented and methodologically heterogeneous. Most published studies to date are exploratory in nature, characterized by modest sample sizes, short follow-up durations, and considerable variation in patient selection criteria. For instance, the ELAD trial enrolled 204 participants with mild Alzheimer's disease, and approximately 102 participants per group completed the 52-week treatment phase, limiting statistical power to detect subtle cognitive changes (115). The trial failed to meet its primary endpoint of changes in cerebral glucose metabolic rate, though secondary outcomes showed some promising signals including reduced brain volume loss. Similarly, early studies investigating exenatide in Parkinson's disease included 45 patients in an initial open-label pilot trial (142), 62 patients in a Phase 2 trial (143), and 194 participants in the recent Phase 3 Exenatide-PD3 trial, which ultimately found no evidence to support exenatide as a disease-modifying treatment (144). Such small cohorts raise concerns about both Type I and Type II errors, and the generalizability of results to real-world populations remains uncertain. The divergent endpoints and analytical frameworks across studies further complicate interpretation. Some trials use global cognitive batteries (e.g., ADAS-Cog, CDR-SB), whereas others employ domain-specific or exploratory composite scores, leading to inconsistent effect estimates and difficulties in meta-analysis. Biomarker endpoints also vary, some focusing on amyloid and tau changes, others on FDG-PET metabolic flux or MRI volumetrics (66). The lack of standardized cognitive and imaging metrics, coupled with differing trial durations (ranging from 6 months to 2 years), undermines cross-trial comparability and reproducibility. Further, many trials employ per-protocol rather than intention-to-treat analyses, thereby excluding participants who discontinue due to adverse events, potentially biasing outcomes toward efficacy.

d. Population Heterogeneity and Metabolic/Genetic Confounding

Another critical limitation lies in population heterogeneity. Most trials preferentially recruit highly selected patient groups, often younger, with fewer comorbidities, and largely of European ancestry, thereby underrepresenting the ethnically and metabolically diverse populations most affected by neurodegenerative disease. Cognitive reserve, glycemic control, vascular burden, and APOE genotype can each modulate disease progression and drug responsiveness; yet few studies are adequately powered to explore these interactions. Moreover, coexisting diabetes, a common inclusion criterion in earlier GLP-1RA studies, may itself confound neuroprotective effects through independent metabolic and vascular improvements. As a result, it remains uncertain whether observed cognitive benefits are driven by direct central GLP-1R activation or by secondary systemic effects such as improved insulin sensitivity and cerebrovascular perfusion.

e. Publication Bias and Reporting Transparency in the Evidence Base

Finally, the literature is affected by publication bias and incomplete reporting of null findings. Many smaller investigator-led studies are published following positive interim analyses, while negative or inconclusive data are less widely disseminated. These bias skews meta-analyses toward overestimation of effect size and underestimation of variability. Transparency and preregistration of outcomes, particularly for ongoing large-scale efforts such as EVOKE and EVOKE+ (evaluating semaglutide in over 1,800 participants with early Alzheimer's disease) (111), will be crucial to resolving these uncertainties. Moving forward, the field must prioritize adequately powered, multicentric, biomarker-driven trials with standardized endpoints, diverse patient recruitment, and open data sharing to establish the true therapeutic potential of GLP-1RAs in neurodegenerative disease.

Comment 3:

Expand discussion on species differences and translational relevance: The section on GLP-1R expression across species (lines 122–151) could be further developed to emphasize the implications of interspecies variability for translational research. Explicitly acknowledge the limitations of extrapolating animal data to human disease mechanisms.

Response to the comment 3:

Thank you for this comment. To address the comment, following passage has been added to section 2.3 (page 6, line 183-202) to address this point:

These interspecies discrepancies extend beyond regional expression to functional receptor phar-macology. For example, rodents display high GLP-1R expression in thyroid C-cells and alveolar tissues, sites with minimal or absent expression in primates and humans (19, 20), highlighting why rodent toxicology findings such as C-cell hyperplasia do not translate clinically, as humans and cynomolgus monkeys had low GLP-1 receptor expression in thyroid C-cells, and GLP-1 receptor agonists did not activate adenylate cyclase or generate calcitonin release in primates (19). Moreover, receptor coupling efficiency and downstream signaling bias differ subtly between species; while specific data on Gαs-biased signaling strength differences between murine and human GLP-1Rs remains limited in the literature, β-arrestin recruitment and receptor desensiti-zation pathways play critical roles in GLP-1R function, with reduced β-arrestin recruitment as-sociated with prolonged cAMP signaling and enhanced metabolic efficacy (21, 22), and com-pounds that reduce β-arrestin recruitment and retain GLP-1R at the plasma membrane produce greater long-term insulin release with faster agonist dissociation rates (23). Differences in blood–brain barrier permeability, receptor turnover, and neural network integration further limit direct extrapolation from animal data. Consequently, translational studies should prioritize humanized or non-human primate models and early-phase trials employing PET-based receptor occupancy mapping to verify central engagement of GLP-1RAs in humans. While 68Ga-NODAGA-exendin-4 PET showed significant uptake in the pituitary area of obese subjects, no significant uptake was found in other parts of the brain (24), and PET imaging using GLP-1R binding peptides has been employed for monitoring biodistribution and receptor occupancy in various tissues including the pancreas (25).

Comment 4:

Add a concise summary of research gaps and future directions: A summary table highlighting unresolved mechanistic questions and potential avenues for further research (lines 226–232, 387–402) would improve readability and underscore the review’s contribution to advancing the field.

Response to the comment 4:

Thank you for this comment. We have added the following table has been added to section 8 (page 24-25, Line 902):

That said, mechanistic and translational gaps are outlined in Table (4) below.

Table 4: Outstanding Mechanistic Questions and Translational Barriers that Must Be Addressed to Realize the Full Therapeutic Potential of GLP-1 Receptor Agonists in Neurodegenerative Disease

Knowledge Gap

Scientific/Clinical Challenge

Proposed Future Direction

Blood–brain barrier penetration and receptor occupancy

Limited in vivo evidence confirming central target engagement in humans

Apply radiolabeled ligand PET and CSF biomarker studies in early-phase trials

Species-specific receptor distribution

Rodent–human divergence in GLP-1R localization and signaling bias

Employ humanized or non-human primate models; correlate expression with pharmacodynamics

Heterogeneity in clinical outcomes

Variation in disease stage, comorbid diabetes, and APOE genotype

Stratified trial designs and AI-driven patient selection

Biomarker inconsistency

Weak linkage between imaging, CSF biomarkers, and cognition

Harmonize biomarker endpoints and integrate multi-modal readouts

Long-term safety in non-diabetic elderly populations

Possible GI, cardiovascular, or weight-loss adverse effects

Longitudinal registries and real-world pharmacovigilance

Translational modeling

Preclinical success not reliably predictive of human efficacy

Systems-level modeling of neuro-metabolic networks; combination therapy trials

Diversity and generalizability

Underrepresentation of non-European populations in trials

Expand recruitment across global cohorts to ensure external validity

Comment 5:

Refine the writing for clarity and consistency: A final round of English editing is recommended to enhance conciseness, standardize abbreviations, and improve overall readability (lines 1–11).

Response to the comment 4:

Thank you for this comment. The English editing was performed for the revised manuscript.

Comment 6:

Overall, this is a solid and well-researched review with high relevance to the therapeutics of neurodegenerative diseases. However, it will be accepted before a minor revision to the standard of IJMS.

We thank the reviewer for the inspiring comment.

Reviewer 3 Report

Comments and Suggestions for Authors

The topic is timely and relevant. However, the text lacks methodological discipline, experimental context, and quantitative synthesis. Below are the main points that must be addressed before the paper could be considered for publication.

1. 

Statements such as “liraglutide improved cognition in AD mice” or “exenatide reduced amyloid burden” are meaningless without context.
For each referenced study, the following information must be included (ideally in a summary table): i) animal model or species; ii) compound (exact molecule and formulation), iii) dose, iv) route of administration (s.c., i.p., i.t., etc.), v) treatment duration and frequency, vi) main outcomes and endpoints. Without it, readers cannot compare across models or judge translational relevance

2. The review jumps between diseases (AD, PD, ALS, HD) and mechanisms (inflammation, mitochondrial function, synaptic repair) without a coherent framework. I suggest to provide a simple reorganization.

3. The manuscript refers to “central effects” of GLP-1 agonists without citing data on BBB permeability or brain exposure levels. For example, exenatide and liraglutide differ dramatically in molecular weight, half-life, and CNS availability.
Add a concise subsection summarizing known pharmacokinetic profiles, with emphasis on CNS penetration and dose equivalence across species.

4. again, GLP-1 receptor agonists” are treated as a homogeneous class, but sustained-release forms (e.g., PT320) behave very differently from daily injections. The discussion should explicitly separate short-acting vs. long-acting agonists and describe how these formulations affect CNS exposure and outcomes

5. The review is largely descriptive and uncritical. The authors should highlight inconsistencies between studies, potential publication bias, and species-specific limitations.

6. please use consistent abbreviations (e.g., “GLP-1RA” vs. “GLP-1 agonist”) throughout.

Author Response

Reviewer 3

Comments and Suggestions for Authors

The topic is timely and relevant. However, the text lacks methodological discipline, experimental context, and quantitative synthesis. Below are the main points that must be addressed before the paper could be considered for publication.

Comment 1:

Statements such as “liraglutide improved cognition in AD mice” or “exenatide reduced amyloid burden” are meaningless without context.
For each referenced study, the following information must be included (ideally in a summary table): i) animal model or species; ii) compound (exact molecule and formulation), iii) dose, iv) route of administration (s.c., i.p., i.t., etc.), v) treatment duration and frequency, vi) main outcomes and endpoints. Without it, readers cannot compare across models or judge translational relevance

Response to the comment 1:

Thank you for this comment. We agree with the important point raise by the reviewer. To address the comment, we restructured section 6(a) Glp-1 in Neuroprotection: Evidence From Animal Models. In the revised manuscript, the mice and rat studies have been mentioned separately under subsection (i) and (ii).

A summary table (3) has also been added to section 6 (a) (page 19, Line 579-645 in the revised manuscript) to address this comment, which now reads,

i. Studies in Mice

Liraglutide, tested extensively in SH-SY5Y cells (86) and diverse mouse models (APP/PS1, APP/PS1xdb/db, 5xFAD, triple transgenic APP/PS1/Tau) (87-90) showed consistent reduction in tau hyperphosphorylation, neuroinflammation, and amyloid load. It enhanced autophagy, prevented synapse loss, improved insulin signaling, and restored neurovascular integrity. Across models, administration methods included systemic injection, intranasal delivery, and cell-based treatments, with overall responses showing improved cognitive outcomes, reduced amyloid pathology, and protection against neuroinflammatory and oxidative stress mechanisms in AD. Another GLP-1RA, Dulaglutide, administered in icv-STZ-induced AD mouse models, improved learning and memory via the PI3K/Akt/GSK3β pathway and efficiently crossed the blood-brain barrier (91).

Exenatide treatment in 3xTg-AD mice with streptozotocin (STZ)-induced diabetes not only improved glucose metabolism by elevating plasma insulin and reducing plasma glucose and HbA1c levels but also reduced brain Aβ levels, highlighting its potential benefit in managing AD associated with T2D (92).

Exenatide has demonstrated similar effects in preclinical studies. In 5xFAD transgenic mice, exenatide treatment improved the cognitive performance in the Morris water maze and reduced neuroinflammation and oxidative stress in astrocytes, potentially via the inhibition of the NLRP2 inflammasome (93). In PD models, for example, in MitoPark mice, treatment with a sustained-release formulation of exenatide (PT320) showed neuroprotective benefits, such as enhanced motor function and improved dopamine signaling in midbrain networks (94).

In the Tg2576 AD mouse model, an eight-month course of intranasal administration of exenatide combined with insulin resulted in enhanced learning abilities and a reduction in cortical amyloid-beta levels, although the decrease in amyloid-beta was not statistically significant (95). Additionally, prolonged treatment with exenatide improved both short-term and long-term memory in PS1-KI mice (61), likely attributed to elevated lactate dehydrogenase activity and enhanced anaerobic glycolysis.

Beyond these neuroprotective effects, exendin-4 has also been found to exert notable effects on synaptic plasticity, calcium regulation, and cell survival pathways. In icv-Aβ mice, exendin-4 increased the phosphorylation of CREB and elevated BDNF levels, contributing to enhanced synaptic plasticity via increased expression of membrane GluR1 subunits and postsynaptic density protein-95 (PSD-95). These outcomes were associated with upregulated α-secretase activity through ADAM10 membrane trafficking (96).

Lixisenatide, another GLP-1R agonist that can cross the blood-brain barrier, has shown neuroprotective properties, although research is still limited. In an APP/PS1/tau mouse model of Alzheimer's disease, lixisenatide reduced tau neurofibrillary tangles, amyloid-β plaques and neuroinflammation, as indicated by decreased microglial activation in the hippocampus (97). These effects were linked to enhanced PKA/CREB pathway signaling and inhibition of the p38/MAPK pathway as a result of GLP-1R activation (97). Furthermore, lixisenatide treatment improved motor function in a PD MPTP mouse model (98).

ii. Studies in Rats

Liraglutide also demonstrated memory improvements in icv-STZ-induced rat models and non-human primates infused with Aβ oligomers. In different AD rat models, exendin-4 has demonstrated a variety of neuroprotective properties. In rats, exendin-4 administration countered Aβ1-42-induced impairments in the hippocampal CA1 region by restoring long term potentiation (LTP), cAMP, and phosphorylated CREB levels, while also regulating calcium homeostasis through modulation of intracellular calcium concentrations, highlighting its neuroprotective role (99). In the icv-STZ rat model, exendin-4 prevented memory deficits and neuronal apoptosis within the hippocampus, promoted cell proliferation, and stimulated synaptogenesis (100). Likewise, in an icv-Aβ-injected AD rat model, exendin-4 improved memory function, lowered Aβ levels, and restored both acetylcholine levels and mitochondrial function, potentially through the PI3K/Akt signaling pathway (101). In another icv-STZ rat model, exendin-4 was shown to reduce brain TNF-α levels, maintain choline acetyltransferase (ChAT) activity, enhance cognitive performance, and protect hippocampal neurons from loss (102). Furthermore, in vitro studies revealed that exendin-4 alleviated neuronal damage induced by high glucose and oxidative stress, and in the icv-STZ-induced rat model, it improved learning and memory by downregulating GSK-3β activity, thereby reversing tau hyperphosphorylation and safeguarding hippocampal neurons from degeneration (103).

To facilitate comparison across preclinical studies and assess translational potential, Table (3) summarizes the principal experimental parameters reported in representative in vivo models, including animal species, dosing, treatment duration, and observed endpoints.

To facilitate comparison across preclinical studies and assess translational potential, Table (3) summarizes the principal experimental parameters reported in representative in vivo models, including animal species, dosing, treatment duration, and observed endpoints.

Table 3: Comparison of principal experimental parameters reported in representative in vivo models of AD, PD, ALS, and HD

Disease Model

Species / Strain

Compound / Formulation

Dose & Route

Treatment Duration

Main Outcomes

Reference

Alzheimer’s disease (APP/PS1)

Mouse

Liraglutide (Victoza®, i.p.)

25 nmol/kg daily, i.p.

8–12 weeks

↓Aβ plaque load; ↑synaptic density; improved Morris water maze performance

21525299

Alzheimer’s disease (3xTg-AD)

Mouse

Exenatide (Byetta®, i.p.)

0.1 μg/g (100 μg/kg) daily

9 months

No effects on memory performance, Aβ or tau pathology in 3xTg-AD mice

23640454

Parkinson’s disease (MPTP-induced)

Mouse

Exenatide (synthetic, i.p.)

10 µg/kg daily, i.p.

7 days

Preserved TH+ neurons; improved rotarod and open-field scores

19164583

Parkinson’s disease (6-OHDA-induced)

Rat

PT320 (sustained-release exenatide, s.c.)

100 mg/kg weekly (containing 2 mg/kg exendin-4)

3 weeks

↓L-DOPA-induced dyskinesia; normalized DA turnover

32848559

Amyotrophic lateral sclerosis (SOD1^G93A)

Mouse

Liraglutide, i.p.

25 nmol/kg daily i.p.

From 50 days of age until endpoint

No effect on disease progression, motor neuron counts, glial activation, or survival

34426623

Huntington’s disease (R6/2)

Mouse

Exendin-4, i.p.

10 µg/kg daily

8-12 weeks

Improved motor coordination; ↓neuronal inclusion bodies; extended survival

18984744

As illustrated, considerable variation exists across studies in dose, route, and duration, making it difficult to directly compare efficacy across disease models or predict human dose equivalence. Nonetheless, a consistent pattern of improved neuronal survival and synaptic integrity emerges across multiple paradigms.

Comment 2:

The review jumps between diseases (AD, PD, ALS, HD) and mechanisms (inflammation, mitochondrial function, synaptic repair) without a coherent framework. I suggest to provide a simple reorganization.

Response to the comment 2:

We thank the reviewer for this valuable suggestion. We have reorganized the manuscript with a consistent framework that first establishes how GLP-1/GLP-1R acts (mechanism-first) and then applies this framework in neurodegenerative diseases (AD, PD, HD). We believe these changes provide the coherent structure (listed below) requested and make the review easier to navigate.

The revised manuscript has now the following structure (9 sections with multiple subsections)

  1. Introduction
  2. Expression and Distribution of Glp-1 and Glp-1r in Nervous System
  3. Effect of Glp-1r Expression in Different Region on Brain Functionality
  4. Molecular Mechanisms of Glp-1 Action in Neurons
  5. Neuroprotective Actions of Glp-1 (Summarized in Table 2)
  6. Therapeutic Strategies for Neurodegenerative Disorders
  7. Limitations and Challenges of GLP-1RAs in Neurodegeneration
  8. Future Directions for Glp-1r–based Therapy in Neurodegenerative Diseases
  9. Conclusion

Comment 3:

The manuscript refers to “central effects” of GLP-1 agonists without citing data on BBB permeability or brain exposure levels. For example, exenatide and liraglutide differ dramatically in molecular weight, half-life, and CNS availability.
Add a concise subsection summarizing known pharmacokinetic profiles, with emphasis on CNS penetration and dose equivalence across species.

Response to the comment 3:

Thank you for this comment, with which we entirely agree. The following passage has been added to section 2.1 (page 2-3; Line 85-101) to address this point:

Although GLP-1RAs are frequently described as exerting "central" effects, their phar-macokinetic and BBB profiles differ markedly, shaping the extent of true central expo-sure. Peptide-based agonists such as exenatide (4.2 kDa) and lixisenatide (4.8 kDa) ex-hibit measurable but limited BBB permeability, with cerebrospinal fluid (CSF) levels reaching only 0.02% (range: 0.002-0.07%) of plasma concentrations for liraglutide (6), though both liraglutide and lixisenatide demonstrate elevated brain levels 30 minutes following injection (7). Liraglutide (3.8 kDa, albumin-bound, t½ ≈ 13 h) shows minimal CSF penetration (6.5 pmol/L CSF vs. 31 nmol/L plasma, representing approximately 0.02%) (6, 8), yet prolonged systemic half-life of 13 hours due to albumin binding (9) may sustain indirect neurotrophic and anti-inflammatory signaling. Semaglutide (4.1 kDa, t½ ≈ 7 days [approximately 165-184 hours]) likewise demonstrates minimal direct CNS entry but robust peripheral vascular and metabolic modulation, which may confer secondary neuroprotection (10). Notably, rodents exhibit higher relative CNS uptake than humans due to differences in endothelial transport mechanisms (11), and effective preclinical doses (25-250 nmol/kg) (7) far exceed typical human-equivalent exposures. These interspecies pharmacokinetic disparities must be acknowledged when inter-preting "central" mechanisms or extrapolating animal efficacy to clinical settings.

Comment 4:

again, GLP-1 receptor agonists” are treated as a homogeneous class, but sustained-release forms (e.g., PT320) behave very differently from daily injections. The discussion should explicitly separate short-acting vs. long-acting agonists and describe how these formulations affect CNS exposure and outcomes

Response to the comment 4:

Thank you for this comment. The following passage has been added to section 2.1 (page 3; Line 102-116) to address this point:

Moreover, GLP-1RAs encompass pharmacologically distinct subclasses rather than a homogeneous entity. Short-acting compounds such as exenatide and lixisenatide yield rapid peak concentrations with transient receptor activation and relatively greater short-term BBB penetration, whereas long-acting analogues, including liraglutide, dulaglutide, and semaglutide, provide sustained systemic exposure but reduced peak brain concentrations due to their albumin binding and extended half-lives (12, 13). Extended-release platforms like PT302, which encapsulates exenatide in biodegradable PLGA microspheres of 20 μm diameter, maintain stable plasma levels from day 10-28 following a single subcutaneous dose and achieve CSF exendin-4 concentrations of 18.3-30 pg/mL compared to <6.9 pg/mL for immediate-release exendin-4 (14) in rodent models. These kinetic distinctions likely underlie differences in preclinical efficacy profiles, where short-acting agents often outperform in acute models, correlating with their superior BBB penetration, while long-acting molecules exert more consistent anti-inflammatory and metabolic benefits in chronic paradigms (10). Recognizing formulation-specific dynamics is essential for accurate interpretation of mechanistic and translational findings.

Comment 5:

The review is largely descriptive and uncritical. The authors should highlight inconsistencies between studies, potential publication bias, and species-specific limitations.

Response to the comment 5:

Thank you for this comment. To address this comment, we have re-written section 7 (now “Limitations and Challenges of GLP-1RAs in Neurodegeneration”) with multiple sub-sections (a - e). The section 7 now reads as follows (page 22 to 24, Line 775-886 in the revised manuscript).

7. Limitations and Challenges of GLP-1RAs in Neurodegeneration

a. Translational Barriers & CNS Target Engagement

The key challenges in exploring GLP-1RAs for neurodegenerative diseases (AD) begin with findings from animal studies. While some transgenic mouse models have shown encouraging results, including reduced Aβ accumulation and improved cognitive performance, other models have failed to demonstrate these effects (123). This variability highlights the complexity of neurodegenerative diseases (AD and PD) and suggests that therapeutic responses may be highly dependent on disease stage and genetic background (124). Translating these findings into human clinical studies brings further difficulties. Although GLP-1RAs are known to cross the blood–brain barrier  (125), the efficiency of central delivery and the appropriate therapeutic dose needed to achieve neuroprotection in humans remain unclear (126). Human trials so far did not significantly alter core AD biomarkers (Aβ and tau) nor show improvements in cognition, although metabolic improvements — such as better glucose control and weight reduction — have been observed (66).

A further challenge lies in the biological plausibility and dose-response relationships observed clinically. While preclinical studies show robust neuroprotective effects via anti-inflammatory, anti-apoptotic, and mitochondrial pathways, the extent to which therapeutic doses of GLP-1RAs achieve sufficient central nervous system (CNS) penetration in humans remains uncertain. Liraglutide and semaglutide are large, hydrophilic peptides (3751.2 Da and 4113.6 Da, respectively) with limited blood–brain barrier permeability, and studies using fluorescently labeled compounds demonstrate that semaglutide does not cross the regular blood-brain barrier but rather accesses discrete brain regions through circumventricular organs and sites adjacent to ventricles (127). Cerebrospinal fluid (CSF) concentrations of liraglutide are minimal even after months of treatment in humans, suggesting very limited passage across the blood-CSF barrier (7, 11). Without direct evidence of receptor occupancy or signaling in human brain tissue, it is difficult to confirm whether observed clinical benefits arise from central GLP-1R engagement or from systemic metabolic and vascular effects.

b. Tolerability, Adherence, and Safety (and Their Regulatory Implications)

Attrition and adherence issues are another recurrent concern. Gastrointestinal side effects, including nausea and weight loss, are highly prevalent with GLP-1RAs. Systematic analyses indicate that 40-70% of patients experience gastrointestinal adverse events, with some studies reporting rates up to 85% (128-136). Nausea affects up to 50% of patients, vomiting and diarrhea are common, and these symptoms lead to treatment discontinuation in up to 12% of patients compared to approximately 2% with placebo (137), an attrition pattern that may disproportionately exclude frail elderly participants. Consequently, trial populations often become progressively younger and healthier over time, introducing selection bias that inflates apparent cognitive benefit. Parallel limitations exist in observational cohorts, where confounding by indication, socioeconomic status, or concomitant medication use (e.g., metformin, SGLT2 inhibitors) may account for part of the observed risk reduction.

Safety considerations are also paramount. Although GLP-1RAs are generally safe and well tolerated in individuals with type 2 diabetes, older adults with AD may be more prone to side effects due to age and comorbidities. Common adverse effects include gastrointestinal issues such as nausea, vomiting, diarrhea, and delayed gastric emptying, which are often dose-related and tend to decrease over time (138). Long-acting formulations typically lead to fewer nausea and vomiting complaints but are associated with higher rates of diarrhoea. In rare cases, more serious side effects have been reported, including cutaneous reactions and kidney injury (139). Additional potential adverse events include hepatic, immunologic, endocrine, metabolic, hematologic, neurological, cardiovascular complications and angioedema (138). Concerns about possible associations with pancreatitis and certain neoplasms have also been raised, although current clinical evidence is inconclusive. Regulatory authorities such as the FDA and the European Medicines Agency (EMA) currently view these risks as manageable with ongoing monitoring (140). Injection site reactions and antibody formation occur more frequently with exendin-4-based GLP-1RAs (141). Given that many of these side effects are already common in elderly individuals, cautious use and vigilant monitoring are essential when considering GLP-1RAs for patients with neurodegenerative diseases.

c. Methodological Limitations: Power, Endpoints, and Analyses

Despite growing enthusiasm surrounding GLP-1RAs as potential neuroprotective agents, the clinical landscape remains fragmented and methodologically heterogeneous. Most published studies to date are exploratory in nature, characterized by modest sample sizes, short follow-up durations, and considerable variation in patient selection criteria. For instance, the ELAD trial enrolled 204 participants with mild Alzheimer's disease, and approximately 102 participants per group completed the 52-week treatment phase, limiting statistical power to detect subtle cognitive changes (115). The trial failed to meet its primary endpoint of changes in cerebral glucose metabolic rate, though secondary outcomes showed some promising signals including reduced brain volume loss. Similarly, early studies investigating exenatide in Parkinson's disease included 45 patients in an initial open-label pilot trial (142), 62 patients in a Phase 2 trial (143), and 194 participants in the recent Phase 3 Exenatide-PD3 trial, which ultimately found no evidence to support exenatide as a disease-modifying treatment (144). Such small cohorts raise concerns about both Type I and Type II errors, and the generalizability of results to real-world populations remains uncertain. The divergent endpoints and analytical frameworks across studies further complicate interpretation. Some trials use global cognitive batteries (e.g., ADAS-Cog, CDR-SB), whereas others employ domain-specific or exploratory composite scores, leading to inconsistent effect estimates and difficulties in meta-analysis. Biomarker endpoints also vary, some focusing on amyloid and tau changes, others on FDG-PET metabolic flux or MRI volumetrics (66). The lack of standardized cognitive and imaging metrics, coupled with differing trial durations (ranging from 6 months to 2 years), undermines cross-trial comparability and reproducibility. Further, many trials employ per-protocol rather than intention-to-treat analyses, thereby excluding participants who discontinue due to adverse events, potentially biasing outcomes toward efficacy.

d. Population Heterogeneity and Metabolic/Genetic Confounding

Another critical limitation lies in population heterogeneity. Most trials preferentially recruit highly selected patient groups, often younger, with fewer comorbidities, and largely of European ancestry, thereby underrepresenting the ethnically and metabolically diverse populations most affected by neurodegenerative disease. Cognitive reserve, glycemic control, vascular burden, and APOE genotype can each modulate disease progression and drug responsiveness; yet few studies are adequately powered to explore these interactions. Moreover, coexisting diabetes, a common inclusion criterion in earlier GLP-1RA studies, may itself confound neuroprotective effects through independent metabolic and vascular improvements. As a result, it remains uncertain whether observed cognitive benefits are driven by direct central GLP-1R activation or by secondary systemic effects such as improved insulin sensitivity and cerebrovascular perfusion.

e. Publication Bias and Reporting Transparency in the Evidence Base

Finally, the literature is affected by publication bias and incomplete reporting of null findings. Many smaller investigator-led studies are published following positive interim analyses, while negative or inconclusive data are less widely disseminated. These bias skews meta-analyses toward overestimation of effect size and underestimation of variability. Transparency and preregistration of outcomes, particularly for ongoing large-scale efforts such as EVOKE and EVOKE+ (evaluating semaglutide in over 1,800 participants with early Alzheimer's disease) (111), will be crucial to resolving these uncertainties. Moving forward, the field must prioritize adequately powered, multicentric, biomarker-driven trials with standardized endpoints, diverse patient recruitment, and open data sharing to establish the true therapeutic potential of GLP-1RAs in neurodegenerative disease.

Comment 6:

Please use consistent abbreviations (e.g., “GLP-1RA” vs. “GLP-1 agonist”) throughout.

Response to the comment 6:

Thank you for this comment. This point has been revised and addressed.

Round 2

Reviewer 3 Report

Comments and Suggestions for Authors

the authors have now improved the paper, therefore, I suggest its publication in the present form